# Unexpected response of nitrogen deposition to nitrogen oxide controls and implications for land carbon sink

Mingxu Liu [1,12], Fang Shang[1,12], Xingjie Lu[2,12], Xin Huang [3], Yu Song [1✉], Bing Liu[4], Qiang Zhang [5], Xuejun Liu [6], Junji Cao[7,8], Tingting Xu[9], Tiantian Wang [1], Zhenying Xu [1], Wen Xu[6], Wenling Liao[1], Ling Kang[1], Xuhui Cai[1], Hongsheng Zhang [10], Yongjiu Dai[2] & Tong Zhu [1,11✉]

Terrestrial ecosystems in China receive the world's largest amount of reactive nitrogen (N) deposition. Recent controls on nitrogen oxides ($NO_x = NO + NO_2$) emissions in China to tackle air pollution are expected to decrease N deposition, yet the observed N deposition fluxes remain almost stagnant. Here we show that the effectiveness of $NO_x$ emission controls for reducing oxidized N ($NO_y = NO_x$ + its oxidation products) deposition is unforeseen in Eastern China, with one-unit reduction in $NO_x$ emission leading to only 55–76% reductions in $NO_y$-N deposition, as opposed to the high effectiveness (around 100%) in both Southern China and the United States. Using an atmospheric chemical transport model, we demonstrate that this unexpected weakened response of N deposition is attributable to the enhanced atmospheric oxidizing capacity by $NO_x$ emissions reductions. The decline in N deposition could bear a penalty on terrestrial carbon sinks and should be taken into account when developing pathways for China's carbon neutrality.

[1] State Key Joint Laboratory of Environmental Simulation and Pollution Control, College of Environmental Sciences and Engineering, Peking University, Beijing 100871, China. [2] School of Atmospheric Sciences, Sun Yat-sen University, Guangzhou 510275, China. [3] Joint International Research Laboratory of Atmospheric and Earth System Sciences, School of Atmospheric Sciences, Nanjing University, 210023 Nanjing, China. [4] Environmental Quality Forecast Center, China National Environmental Monitoring Center, 100012 Beijing, China. [5] Ministry of Education Key Laboratory for Earth System Modeling, Center for Earth System Science, Institute for Global Change Studies, Tsinghua University, 100084 Beijing, China. [6] Key Laboratory of Plant-Soil Interactions of MOE, Beijing Key Laboratory of Farmland Soil Pollution Prevention and Remediation, College of Resources and Environmental Sciences, National Academy of Agriculture Green Development, China Agricultural University, Beijing 100193, China. [7] State Key Laboratory of Loess and Quaternary Geology, Institute of Earth Environment, Chinese Academy of Sciences, 710061 Xi'an, China. [8] Key Laboratory of Aerosol Chemistry and Physics, Institute of Earth Environment, Chinese Academy of Sciences, 710061 Xi'an, China. [9] Department of Environmental Science and Engineering, College of Ecology and Environment, Chengdu University of Technology, Chengdu 610059, China. [10] Laboratory for Atmosphere-Ocean Studies, Department of Atmospheric and Oceanic Science, School of Physics, Peking University, 100871 Beijing, China. [11] Beijing Innovation Center for Engineering Science and Advanced Technology, Peking University, Beijing 100871, China. [12] These authors contributed equally: Mingxu Liu, Fang Shang, Xingjie Lu. ✉email: songyu@pku.edu.cn; tzhu@pku.edu.cn

Nitrogen (N) is an essential nutrient element for terrestrial ecosystems but the excess deposition of atmospheric N causes a variety of detrimental environmental impacts, including decreased biological diversity, soil acidification, and soil and water eutrophication[1–4]. Over the last century, increased anthropogenic emissions of major reactive nitrogen, i.e., nitrogen oxides ($NO_x = NO + NO_2$) from fossil fuel combustion and ammonia ($NH_3$) from agricultural production, enhanced the global N deposition[5–7].

China is the world's largest contributor of atmospheric N deposition because of the huge emissions of $NO_x$ (~10.3 Tg N yr$^{-1}$) and $NH_3$ (~8.2 Tg N yr$^{-1}$) from anthropogenic activities[8–10], both exceeding the sum of those in the United States and European Union[11]. Recent modeling and observational evidence suggest that high atmospheric N deposition in China plays an important role in determining terrestrial carbon balance[12,13]. The contribution of ecosystem carbon sequestration is essential for China to achieve the carbon neutrality goal before the year 2060[14], which has recently been announced by the Chinese government to tackle climate change. On the other hand, excess N inputs to the land cause detrimental effects to some terrestrial and aquatic ecosystems[15,16]. In the period of 2010s, the $NH_3$ emission in China has been nearly constant, while the $NO_x$ emission has markedly declined by more than 20% owing to the stringent clean-air actions[9]. However, the observed N deposition fluxes over China did not show a significant decrease as expected and tend to be stabilized in this period[17]. We found that the decline tendency in N deposition fluxes observed in Eastern China appears much lower than the reductions of $NO_x$ emissions (Fig. 1). In a similar manner, particulate nitrate concentrations in air measured in this region remain high or even slightly increase despite those $NO_x$ reductions[18]. By contrast, the N deposition over the United States has been found to respond proportionately to $NO_x$ emission abatement[19]. These phenomena pose a dilemma in how the N deposition over China responds to the recent reductions in $NO_x$ emissions.

N deposition is comprised of dry and wet deposition of oxidized N ($NO_y$, the sum of $NO_x$, mineral $NO_3$-N as gas and aerosol nitric acids and nitrogen pentoxide ($N_2O_5$), and organic nitrates) and reduced N ($NH_x$ = ammonia [$NH_3$] + ammonium [$NH_4^+$]). The daytime photo-oxidation of $NO_2$ by OH radical and the nighttime $NO_3$ radical-involved chemistry are the major pathways for the formation of $NO_3$-N compounds in the lower troposphere. Volatile organic compounds (VOCs) can readily react with OH radicals and promote $O_3$ formation. The heavy $O_3$ pollution and its nonlinear response to $NO_x$ mixing ratios in China are found to be regionally diverse[20–22]. Moreover, $NH_x$-N deposition is likely influenced by variations of nitric acid concentrations that determine the thermodynamic partitioning of ammonia between gas and particulate phases in the atmosphere. Therefore, it is plausible that the recent reductions in $NO_x$ emissions would modulate the atmospheric oxidation capacity and subsequently influence regional N deposition.

To elucidate the comprehensive impacts of ongoing $NO_x$ emission abatement on controlling N deposition in China, particularly on $NO_y$-N, we combined the Weather Research and Forest model with chemistry (WRF-Chem) and a nationwide N deposition measurement dataset[23]. Our results will be useful to evaluate the response of atmospheric N deposition to human activities in the future and to project their impacts on terrestrial ecosystems, especially on the natural carbon sink in China.

## Results

**Characteristics of atmospheric N deposition over China.** We first evaluate the fidelity of our model in simulating N deposition and their precursors over China for the year 2015 (referred to as

Baseline case hereafter; see Supplementary Table 1 for all simulation experiments), using the national N deposition observation network and other in situ atmospheric measurements (Supplementary Text 1, Figs. 1, 2, and Table 2). The model results generally reproduce the observed spatial pattern and magnitude of $NO_y$-N deposition and identify the typical hotspots (10–20 kg N ha$^{-1}$ yr$^{-1}$) of $NO_y$-N deposition fluxes mainly distributed in Eastern China and Southern China, reflecting their high anthropogenic $NO_x$ emission rates. The atmospheric transport of $NO_x$ emissions between those source regions critically determines the regional N deposition. By performing multiple parallel simulations with the anthropogenic emissions excluded in the targeted source regions (see Methods), we find that the local emissions in Eastern China dominate (57%) the $NO_y$–N deposited there, with the remaining part (43%) from the other domestic regions; while $NO_y$-N deposition in Southern China is contributed largely by the $NO_x$ emissions in the surrounding regions (67%). Overall, the annual sum of $NO_y$-N deposition over the Chinese terrestrial land is estimated to be 5.2 Tg N yr$^{-1}$ in 2015, accounting for 47% of the total N deposition, with the remaining part from $NH_x$-N (5.8 Tg N yr$^{-1}$). The dry and wet forms contribute almost equally to $NO_y$-N deposition over Chinese terrestrial land. These modeling results are in good agreement with the multiple-model ensemble mean values in model comparison studies like MICS-Asia III[24]. The ratio of $NO_y$ deposition to the domestic $NO_x$ emission over China is 0.50 in our simulation, also close to another modeling study (0.53)[25].

Specifically, $NO_3$-N (the sum of gas and aerosol $NO_3^-$ and $N_2O_5$) contributes to 83% of $NO_y$-N deposition, followed by $NO_x$ (14%) (Supplementary Table 3). The dominant role of $NO_3$-N compounds in determining $NO_y$-N deposition is attributable to the efficient production of nitric acids (hereafter including both gas and aerosol phases unless noted otherwise) in the polluted atmosphere after the emission of $NO_x$. As reflected by both measurements and modeling results, the annual mean PM$_{2.5}$ (particles with an aerodynamic diameter less than 2.5 µm) nitrate concentrations at near-surface air generally exceed 10 µg m$^{-3}$ over Eastern China (Supplementary Fig. 2). The secondary formation of nitric acids in the lower troposphere mainly involves OH· + $NO_2$ oxidation in the daytime and the $NO_3$· + $NO_2$ reaction at nighttime, which are associated with the active $NO_x$-VOCs-$O_3$ photochemistry[18,26]. Our simulations show that the gas-phase oxidation of $NO_2$ by OH radical and nighttime reactions mediated by $NO_3$ radical are two major pathways of nitric acid formation in wintertime with the relative contributions of 40% and 60%, respectively, while the daytime photochemistry pathway is dominant (90%) in summer. These results support that the response of $NO_y$-N to $NO_x$ emission reductions is controlled by changes in nitric acid production and subsequent concentrations in the air.

The observed annual $NO_y$ deposition fluxes at most of the in situ stations fluctuated by 10% during the period of 2011–2015 despite the notable declines (about 20%) in $NO_x$ emissions and observed $NO_2$ column concentrations in air (Fig. 1a, b). This phenomenon could be partially associated with meteorological conditions (e.g., monsoon, precipitation, and atmospheric diffusion). Our simulations with constant emissions but varying meteorological conditions from 2011 to 2015 yield concurrent positive and negative changes in N deposition fluxes (within −6 and +28%) at those different stations (Supplementary Table 4). For some stations in Eastern China like QZ and ZZ, the meteorological contributions (+15 and −2%), however, cannot fully explain the stabilization or enhancement of $NO_y$ deposition fluxes under the $NO_x$ emission reductions (Fig. 1a). This study proposes that the enhanced atmospheric oxidation capacity by $NO_x$ reductions is a key factor in driving $NO_y$-N deposition variations, as demonstrated in the following.

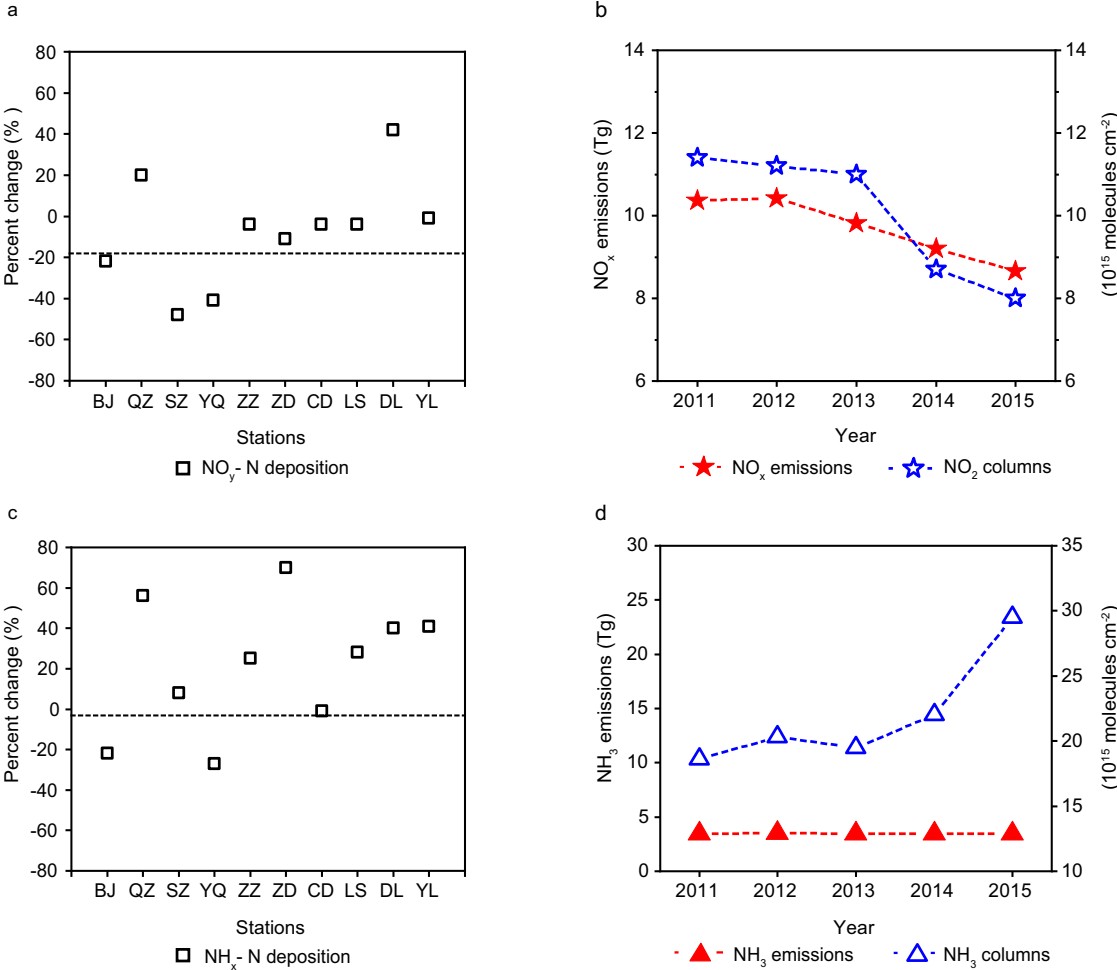

**Fig. 1 The inter-annual variations of observed reactive N deposition fluxes and the precursor emissions and column concentrations in Eastern China during 2011–2015. a** The temporal changes (%) of observed annual $NO_y$-N deposition fluxes at each of the ten stations in Eastern China (the locations are marked in Fig. 2) from 2011 to 2015; the horizontal dashed line (−18%) denotes the tendency of annual $NO_x$ emission in Eastern China from 2011 to 2015. **b** The inter-annual $NO_x$ emissions and observed region-averaged $NO_2$ columns for Eastern China. **c** The percent changes of $NH_x$-N deposition fluxes from 2011 to 2015 at observation sites; the dashed line denotes the tendency of regional $NH_3$ emission. **d** The inter-annual $NH_3$ emissions and observed region-averaged $NH_3$ columns for Eastern China. The deposition observations were derived from ref. [24]. The ten observational stations are Beijing (BJ), Quzhou (QZ), Shangzhuang (SZ), Yangqu (YQ), Zhengzhou (ZZ), Zhumadian (ZD), Changdao (CD), Lingshandao (LS), Dalian (DL), and Yangling (YL). The $NO_x$ emissions were derived from ref. [9]. The $NH_x$-N emissions and column concentrations were derived from ref. [29].

**Response of N deposition to $NO_x$ emission reductions**. In order to ascertain the response of N deposition to $NO_x$ emission controls over China, we performed three parallel simulation experiments by separately imposing 10, 30, and 50% reduction (referred to as RED10, RED30, and RED50 cases) of $NO_x$ emissions relative to the Baseline case (using the emission and meteorological years for 2015) and keeping emissions of other species constant. Figure 2 presents the changes (%) of $NO_y$-N deposition in different forms due to the three emission scenarios. To indicate the relative response of $NO_y$-N deposition to $NO_x$ emission abatement, we calculated the ratios of percentage changes in regionally aggregated deposition fluxes to changes in $NO_x$ emissions[19] (Fig. 2c). The percent changes in national total $NO_y$-N deposition are generally in line with the 10–50% reductions of $NO_x$ emissions but the relative responses are less than 100% in all cases (84% for RED10, 89% for RED30, and 95% for RED50).

The responses of $NO_y$-N deposition fluxes to $NO_x$ emission reductions are spatially diverse between Eastern China and Southern China (Fig. 2a, b and Supplementary Fig. 3). For Eastern China, the 10–50% reductions of $NO_x$ emissions result in only 5.5–38% reductions in $NO_y$-N deposition, corresponding to

the relative responses of 55–76%. In contrast, the reductions in $NO_y$-N deposition in Southern China show the relative responses of 100–110% to $NO_x$ emission reductions. We also examined the sensitivity of emission-deposition responses to inter-annual changes in meteorological fields (e.g., precipitation and monsoon) by comparing the results for 2015 with those simulations using meteorological fields for 2011. The sensitivity simulations show the percent changes of $NO_y$ deposition of 17% in Eastern China and 26% in Southern China due to the 30% $NO_x$ reduction, close to the results for 2015. These results suggest that the current $NO_x$ emission regulation can efficiently reduce $NO_y$-N deposition in Southern China, but not in Eastern China.

Our results further reveal that this unexpected weakened response of $NO_y$-N deposition to $NO_x$ emission abatement in Eastern China is attributable primarily to variations in $NO_3$-N dry deposition. The region features great reductions (by 10–55% for the RED10, RED30, and RED50 cases) in gas-phase $NO_x$ deposition but considerably smaller reductions in $NO_3$-N deposition (−4, −15, and −33%, respectively). Specifically, the changes in $NO_3$-N dry deposition are as low as −3.0% in the RED30 simulation and even positive (+0.3%) in the RED10

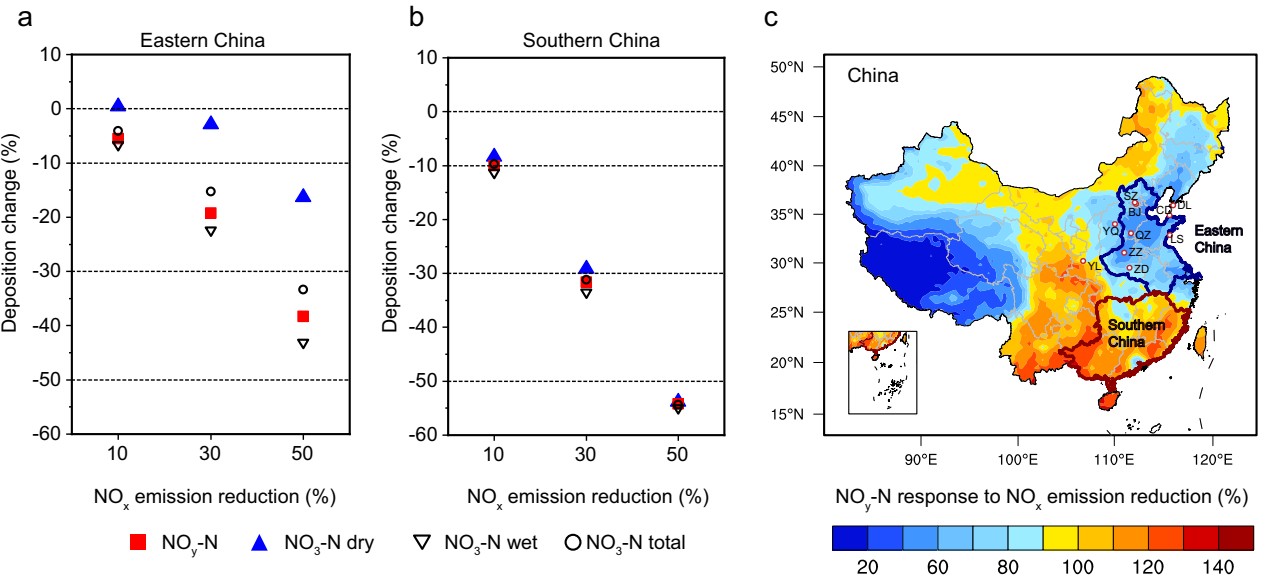

**Fig. 2 Response of NO$_y$-N deposition to 10, 30, and 50% reductions of NO$_x$ emission. a** The percent changes of NO$_y$-N (NO$_x$ and its oxidation products) deposition, NO$_3$-N (nitric acid and nitrogen pentoxide) dry deposition, NO$_3$-N wet deposition, and NO$_3$-N total deposition in Eastern China. **b** The percent changes of NO$_y$-N deposition forms in Southern China. **c** Spatial distribution of the relative response of NO$_y$-N deposition to the 30% NO$_x$ emission reduction. The dots in panel **c** denote the ten observation stations for N deposition fluxes during 2011–2015. Note that values of the relative responses less than 100% in **c** represent the percent reductions of NO$_y$ deposition lower than the reductions of NO$_x$ emission. The map of China was reproduced from the National Geographic Information Resource Directory Service System (https://github.com/huangynj/NCL-Chinamap.git and https://www.webmap.cn/commres.do?method=result100W).

(Fig. 2a). Because the dry form accounts for 41% of total NO$_3$-N deposition, these very few changes in NO$_3$-N dry deposition consequently relieve reductions in NO$_y$-N deposition. Seasonally, we find increases of NO$_3$-N dry deposition are especially notable during fall and winter months (Supplementary Figs. 4, 5, 6), with the seasonal NO$_3$-N dry deposition elevated by +15% during wintertime in the RED30 case. Unlike that, NO$_3$-N dry deposition in Southern China decreases proportionately with NO$_x$ emission reductions. The relative responses of annual NO$_3$-N dry deposition in those three cases range from 62 to 85% for China, from 84 to 108% for Southern China, and from −4 to 33% for Eastern China (Supplementary Fig. 7). This spatially heterogeneous response relationship between the deposition and emissions implies the underlying region-dependent mechanisms, as shown in the following analysis.

The regional N deposition responses to the NO$_x$ reductions are explained by the changes in atmospheric nitric acid (gas + aerosol phases) concentrations. Our simulations reveal increases of nitric acid near-surface concentrations by 6–11% over Eastern China during different seasons (except summertime), even though NO$_x$ emissions decrease by 30% (Fig. 3a). The enhancements of nitric acid production due to the NO$_x$ reduction are most evident during fall and winter periods. These increases in nitric acid are allocated almost equally in the gas-phase and aerosol phase because free ammonia in Eastern China is not sufficient enough to neutralize all gas-phase nitric acid in the air. The particulate fraction of total nitrate (ratio of aerosol nitrate to the total) decreases from 86.1 to 83.5% in Eastern China during wintertime, as a result of a larger increase in gas-phase nitric acid concentration compared to the aerosol form. This is supported by previous modeling evidence that elevated ammonia availabilities would enhance the conversion of nitric acid from gas to aerosol phases in this region[27,28]. On the contrary, the nitric acid concentrations over Southern China are reduced by 15–31% seasonally.

The promoted conversion of NO$_x$ to gaseous nitric acid increases the relative fraction of the latter in the total NO$_y$ species

and subsequently enhances the total NO$_y$ deposition. For example, the average ratio of NO$_2$ to gaseous nitric acid concentrations (in ppb) in the surface layer shifts from 40 in the Baseline case to 25 in the RED30 over Eastern China during winter. Because the dry deposition velocity of gaseous nitric acid (1.5 ± 0.5 cm s$^{-1}$ on the annual average) is much higher than that of NO$_2$ (0.14 ± 0.1 cm s$^{-1}$) in this region, the overall NO$_y$ deposition rate would be enhanced. As demonstrated earlier, the NO$_y$ deposition decreases only by 21% in Eastern China under the 30% NO$_x$ emission reduction, while the NO$_y$ mean column concentration decreases by 33%. Consequently, the overall NO$_y$ lifetime that is estimated using the ratio of its column budget to the deposition shifts from 4.5 days to 4.1 days in Eastern China, indicative of increased NO$_y$ loss through dry deposition. Compared to the Baseline case, more NO$_y$ compounds as NO$_3$-N are deposited into the land with the ratio of NO$_y$ deposition to the regional emission in Eastern China increasing by 16%. Therefore, the enhancement of NO$_y$ deposition rates weakens the response of NO$_y$-N deposition to NO$_x$ emission reductions and explains why the effectiveness of reducing total NO$_y$-N deposition per unit of NO$_x$ emission control is appreciably lower than an ideal 100% value in this region.

We further demonstrate that the enhanced formation of nitric acids over Eastern China is triggered by the nonlinear relationship between NO$_x$ and atmospheric oxidants, e.g., O$_3$ and NO$_3$ radical. Compared to the Baseline simulation, the surface O$_3$ concentrations in the RED30 case show an enhancement throughout the year (Fig. 3b). The averaged increases of O$_3$ mixing ratios range from 11 ppb (equivalent to 27% of the Baseline O$_3$) during summer to 5.6 ppb (46%) during winter. The effect that NO$_x$ reductions inhibit the NO-titration process can favor the accumulation of O$_3$ in the atmospheric boundary layer[29]. The elevated ozone mixing ratios are accompanied by the substantial increases of both OH radical and NO$_3$ radical mixing ratios by more than 100% in this case (Fig. 3c, d), indicating the

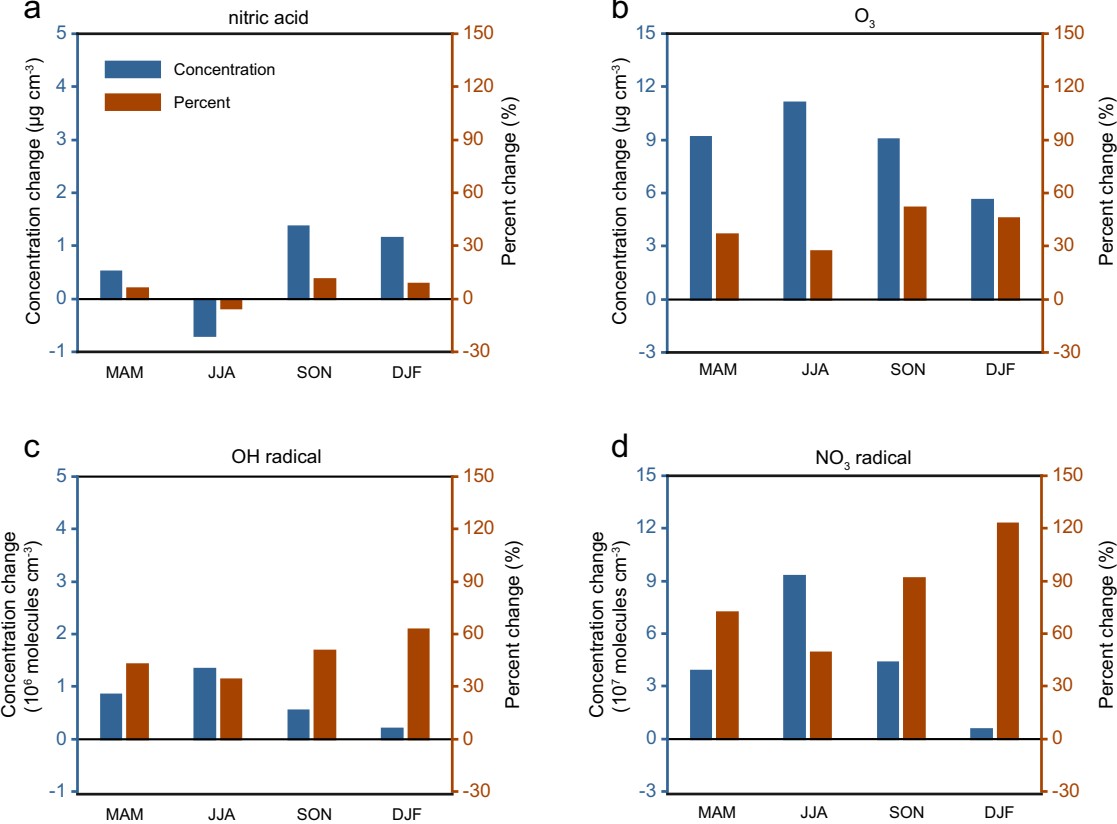

**Fig. 3 Seasonal mean changes in near-surface nitric acid, $O_3$, $NO_3$ radical, and OH radical concentrations in Eastern China caused by a 30% reduction of $NO_x$ emission.** Both absolute and percent changes were calculated for **a** nitric acid concentrations, **b** $O_3$ concentrations, **c** OH radical concentrations, and **d** $NO_3$ radical concentrations. Results in four seasonal periods are shown here, i.e., MAM (March, April, and May), JJA (June, July, and August; local summer), SON (September, October, and November), and DJF (December, January, and February; local winter). The corresponding changes for Southern China are shown in Supplementary Fig. 8.

totally enhanced atmospheric oxidation capacity in both the daytime and nighttime. Consequently, nitric acid concentrations are increased as a result of the enhanced oxidation of $NO_x$ by OH radicals in the daytime and the formation of dinitrogen pentoxide in the nighttime. The enhancement of nitric acid formation by the $NO_x$ reductions is prominent during the fall and winter when $O_3$ formation in Eastern China is probably controlled by strong $NO_x$-saturated conditions and therefore sensitive to reductions of $NO_x$ emissions[22]. The sensitivity simulations (Supplementary Table 1) to identify the relative importance of $NO_x$ removal by daytime and nighttime pathways show that nighttime reactions governed by $NO_3$ radical contributed to 57% of increased production of nitric acid during wintertime, with the remaining part from the daytime photochemical reactions. For Southern China (Supplementary Fig. 8), the atmospheric oxidants (OH and $NO_3$ radical) concentrations are unchanged or decreased, distinct from those in Eastern China.

The contrasting chemical regimes in $O_3$ formation between Eastern and Southern China are rooted in the regional emissions of VOCs and $NO_x$, which could be reflected by the ratio of formaldehyde (HCHO) to $NO_2$ column mixing ratios (e.g., HCHO/$NO_2$) in air. The ratio of around 1.0 has been shown as the threshold between the $NO_x$-saturated (<1) and $NO_x$-limited (>1) regimes for $O_3$ formation[22,30,31]. During summer, the regional mean HCHO/$NO_2$ ratio is 1.2 in Southern China, while as low as 0.24 in Eastern China, indicative of the strong $NO_x$-saturated regime in the latter. During winter, the ratio in Southern China is also significantly higher than in Eastern China (0.38 vs. 0.13). The much lower HCHO/$NO_2$ ratios and the

stronger nonlinear relationship between $NO_x$ reductions and oxidant enhancement in Eastern China are primarily because of the higher emission ratios of $NO_x$ to VOCs and resultant $NO_x$-saturated regime for ozone formation throughout the year. It is well known that Eastern China features the highest $NO_x$ emission rates nationally, while Southern China is subject to greater solar radiation fluxes, higher temperature, and more intensive biogenic VOCs emissions, which jointly lead to the quite different $O_3$-$NO_x$-VOCs photochemistry regimes between the two regions[32,33]. Our modeling results suggest that these different $NO_x$-VOCs-$O_3$ photochemistry regimes account for the regional divergence in $NO_y$-N deposition variations in China.

The reductions of $NO_y$-N wet deposition by decreased $NO_x$ emissions are more efficient than those of $NO_y$-N dry deposition (Fig. 2 and Supplementary Fig. 9). $NO_3$-N wet deposition fluxes could be strongly determined by the nitric acid concentrations aloft that would be efficiently scavenged by cloud and falling precipitation and delivered to Earth's surface. Our modeling results show a clear vertical gradient in the response of ambient nitric acid concentration to $NO_x$ reductions from the surface to the upper layers (about 5 km above ground) (Supplementary Figs. 10 and 11). Unlike the significant enhancement of oxidation capacity and associated nitric acid formation at near-surface levels (except during summer), nitric acid concentrations at higher altitudes (mostly at 1.3–4 km) significantly decreased by around 30% over Eastern China (Supplementary Fig. 10), consistent with the 30% $NO_x$ emission reduction. The percentage increases of $O_3$ and $NO_3$ radical also diminish progressively with altitude and a tipping point for decreasing concentrations occurs

at 1.3–2.5 km above ground. The results imply that the close linkage between $NO_x$ reductions and enhancement of oxidation capacity primarily exists at near-surface levels, where the NO-titration effect could be much more pronounced than that at higher altitudes over Eastern China[34]. The considerable decrease of nitric acid concentrations at high altitudes above 1 km results in substantial reductions (~30%) of $NO_y$-N wet deposition in the region. Southern China shows significant decreases (20–40%) in nitric acid concentrations at both the surface and higher altitudes and the associated change in $NO_y$-N wet deposition is consistent with the $NO_x$ reduction (30%) (Supplementary Fig. 11).

In addition, the aforementioned changes in nitric acids would be accompanied by the phase partitioning of ammonium nitrate and the resultant variations in atmospheric ammonia concentrations and $NH_x$-N deposition. The response of $NH_x$-N deposition to the 30% reduction in $NO_x$ emissions is evaluated here. Over Eastern China, both the modeled annual $NH_x$-N wet deposition and dry deposition decreased slightly (by 0.64 and 1.0%), while for Southern China, the $NH_x$-N dry deposition increased by 2.0% and the wet deposition decreased by 0.88%. The considerable decrease of atmospheric nitric acid concentrations in Southern China results in an increase of gaseous ammonia concentrations by 8.0% and consequently enhances ammonia dry deposition. The changes in the $NH_x$-N deposition over Eastern China and Southern China are −0.80 and −0.13%, respectively, and their contributions to the total N deposition changes are overall negligible compared to $NO_y$-N deposition.

Recently, simultaneous control of VOCs emissions has been proposed to effectively lower the oxidation levels due to their significant contribution to the formation of $O_3$ and OH radicals in the atmosphere[35]. To indicate the future evolution of N deposition with air pollution control measures, we performed another simulation experiment by following one projected anthropogenic emission pathway for 2050 (see Methods), in which $NO_x$ and VOCs emissions are estimated to be reduced by 48 and 34% relative to 2015, respectively. Compared to the RED50 case with a 50% $NO_x$ emission reduction only, the added VOCs emissions control can enhance the efficacy of reducing $NO_y$-N deposition over Eastern China (Supplementary Fig. 12). The associated reduction in the annual mean near-surface nitric acid concentrations is 17%, larger than 3.3% in the RED50 scenario. Consequently, the percentage change in $NO_3$-N dry deposition shifts from −16 to −24% with additional VOCs controls. We attribute those changes to the effectively decreased oxidation capacity owing to VOCs emission abatement, which limits the formation of $O_3$ and partially counteracts the enhancement of oxidants induced by $NO_x$ reductions. Thus, the response of $NO_y$-N deposition to $NO_x$ emissions in the future will be influenced by the coordinated control strategies for $NO_x$ and VOCs emissions.

**Effects of $NO_x$ emission controls on land carbon sink.** During 1980–2010, increased N deposition fluxes significantly enhanced the net carbon sink of the terrestrial land ecosystem in China[12]. However, the substantial reductions of anthropogenic $NO_x$ emissions aiming to control air pollution would in turn dampen the Chinese land carbon sink. Herein, we estimated how the declining N deposition reductions due to N emission controls influence land carbon sink in China, by integrating terrestrial ecosystem model simulations with different N deposition scenarios (see Methods).

Net ecosystem production (NEP), as an indicator of carbon sink, was calculated for the Base and RED30 cases, in which the 30% reduction of $NO_x$ emissions decreases the total N deposition to soils by 1.4 Tg N yr$^{-1}$. This decline in N deposition weakens the plant N uptake for growth and reduces the net primary production (NPP) due to the limitation of nitrogen, which consequently yields a reduction of NEP by 11.2 Tg C yr$^{-1}$ (Fig. 4). The reductions are widely spread with a significant gradient from northwestern China to southeastern China. Noticeably, the regional mean NEP reduction induced by the N deposition change is more significant in Southern China (3.3 g C m$^{-2}$ yr$^{-1}$) than that in Eastern China (0.61 g C m$^{-2}$ yr$^{-1}$), in part because the plant growth in Southern China is more likely limited by the soil nitrogen supply[36,37].

The unexpected weakened response of N deposition to $NO_x$ emission control would revise the regional NEP. As aforementioned, the 30% $NO_x$ emissions reduction results in only a 20% decrease of $NO_y$ deposition in Eastern China, with a relative response of 65% for $NO_y$ deposition. This reduction of atmospheric N inputs weakens both NPP and heterotrophic respiration ($R_h$), yielding a change of NEP by −0.46 Tg C yr$^{-1}$ relative to the Baseline simulation with full N deposition. By contrast, when we applied an ideal (100%) response of $NO_y$ deposition to the 30% $NO_x$ emission reduction, which points to a greater reduction (30%) of N inputs to ecosystems, the resulting change of the NEP can be much larger (−1.80 Tg C yr$^{-1}$) in Eastern China. The results suggest that the nonlinear response of N deposition fluxes to declining $NO_x$ emissions can feedback to ecosystem carbon sinks. Elucidating the dependence of N deposition on N emission variations is thus important for projecting future carbon sequestration potential in terrestrial ecosystems.

## Discussion

This study demonstrates the importance of active $O_3$-VOCs-$NO_x$ photochemistry in modulating $NO_y$-N deposition in China and thus the total N deposition. We find that one-unit control of $NO_x$ emission results in only 55–76% reduction of $NO_y$-N deposition in Eastern China (100–110% in Southern China) because of markedly enhanced atmospheric oxidation capacity and resultant increases in nitric acids. The enhancement of near-surface $O_3$ and $NO_3$ radical occurs even under a 50% reduction in $NO_x$ emissions over Eastern China because VOCs emissions remain high and contribute a lot to the strong oxidation capacity[26]. One similar study focusing on the responses of reactive N deposition to N emissions for the United States has shown the high effectiveness (80–120%) of $NO_x$ emission abatement in reducing $NO_y$-N deposition[19]. The distinct responses between China and the United States reflect their differences in photochemical regimes and hence atmospheric oxidation capacities, which have been observed to be generally higher in China due to intensive anthropogenic emissions[38].

These results could advance the understanding of how the regulations of anthropogenic N emissions impact the N balance in terrestrial ecosystems through atmospheric deposition. By coupling N deposition into a terrestrial ecosystem model, we find that the 30% reduction of $NO_x$ emissions, corresponding to a 12% reduction of total N deposition from domestic sources, decreases the net ecosystem carbon sink by 11.2 Tg C yr$^{-1}$ (1.6%) over China's terrestrial land relative to the baseline case for the year of 2015. Regional decreases in carbon sinks are most notable in southern China, because of the nitrogen-limited feature for plant growth. Similarly, Liu et al.[36] reveal a reduction of gross primary production in southern China by around 5% due to nitrogen limitation; Zhao et al.[37] suggest marked contributions (up to 30%) of anthropogenic nitrogen deposition to NPP and leaf area index in southern China. Our results also show that the nonlinear and region-dependent responses of N deposition to anthropogenic $NO_x$ emission variations modulate net ecosystem production.

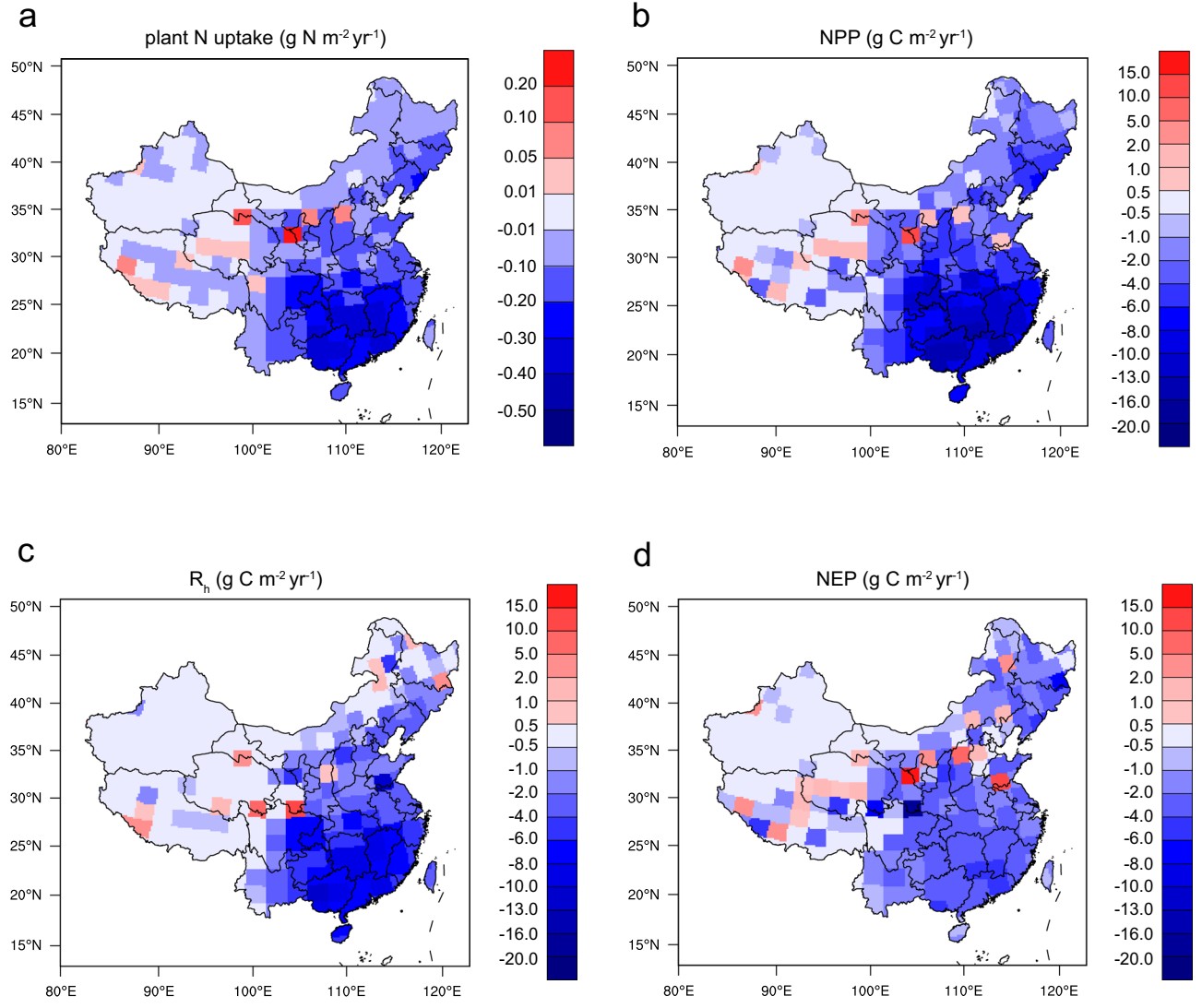

**Fig. 4 Changes in ecosystem carbon balance caused by a 30% reduction of NO$_x$ emission.** Four variables associated with plant growth and carbon sequestration are provided by a land ecosystem model (CLM5): **a** plant N uptake, **b** net primary production (NPP), **c** heterotrophic respiration (R$_h$), and **d** net ecosystem production (NEP). The map of China was reproduced from the National Geographic Information Resource Directory Service System (https://github.com/huangynj/NCL-Chinamap.git and https://www.webmap.cn/commres.do?method=result100W).

Deeper controls on anthropogenic emissions from fossil fuel combustion and agricultural production are desirable in the future to improve air quality and achieve the carbon-neutral goal for China. However, it is likely that those control policies can in turn dampen the terrestrial land carbon sink because of substantially diminished N enrichment. By assuming a linear response of net ecosystem carbon sink to varying N deposition, we estimate natural carbon sequestration of about 93 Tg C yr$^{-1}$ induced by the total anthropogenic N deposition, which accounts for 7–21% of projected land carbon sinks required to reach the carbon neutrality in China[39,40]. Our results suggest that the ongoing N emission controls to tackle air pollution bear a penalty on land carbon sinks, which should be taken into account when developing pathways for China's carbon neutrality before 2060.

The complex biosphere-atmosphere interactions and feedbacks mediate the response of ecosystems to future changes in anthropogenic emissions and resulting atmospheric nitrogen deposition in China. First, the projected reductions in China's NO$_x$ and VOCs emissions would mitigate surface ozone pollution and their damage to plants and primary production[12,41]. Hence, the enhancement of land carbon sink driven by decreasing ozone

concentrations may partially offset the N-control penalty. Moreover, varying nitrogen deposition fluxes likely alter ozone pollution through their modifications to biogenic VOCs emissions, ozone dry deposition velocities, and soil NO$_x$ emissions. These effects have been found to be different between Eastern China and Southern China[37], mainly attributable to the divergent chemical regimes of ozone formation, i.e., NO$_x$-saturated and NO$_x$-limited, as analyzed in this study. A two-way biosphere-atmosphere coupling (e.g., ref. [42]) could drive a more realistic understanding of how China's rapid emission variations modify plant productivity and terrestrial carbon balance. These feedback processes parameterized in models are still subject to considerable uncertainties, which call for integrated long-term observations of air pollution–terrestrial interactions[43].

In addition, some modeling and observational evidence has shown that the large N deposition to the land results in the exceedance of the N critical loads in terrestrial ecosystems[10,44], above which harmful effects including eutrophication and acidification on ecosystems likely occur. These potential exceedances are found mainly in Southern China. Atmospheric N deposition originating from human activities also acts as an important driver

of the N cycle in China's inland waters[15]. For example, Taihu Lake, the third largest freshwater lake in China, has experienced a worsening in water quality due to its high N loadings; inferred from long-term measurements, about one-third of the water N are contributed by N deposition from industries and agricultural production[45]. Hence, the future projected reductions in anthropogenic emissions of both reduced and oxidized nitrogen in China would mitigate the risks of excess N inputs on ecosystems, particularly in Southern China, with the high efficiency of $NO_x$ controls in reducing N deposition. More assessments are warranted to pinpoint the comprehensive impacts of N deposition reduction on regional ecosystems.

While our analysis is confined to the land ecosystems, N deposition in oceans near continental sources may also be affected. Because the enhanced nitric acid production rates due to the $NO_x$ reductions relieve the reduction in N deposition in the source regions, those deposited to open oceans would therefore be reduced more efficiently (the effectiveness >100%) to conserve the mass budgets of nitrogen between the global-scale emission and deposition. It may therefore be important to further investigate the impacts of Chinese $NO_x$ emission abatements on the atmospheric N deposition fluxes in the continental outflow areas like the East China Sea and western Pacific Oceans, where the marine phytoplankton productivity and associated carbon sequestration is likely susceptive to N inputs[46,47].

Due to scarce measurements of $NO_y$-N dry deposition fluxes over China, recent studies usually used satellite-observed $NO_2$ column concentrations to derive the inter-annual trend in $NO_y$-N deposition by assuming a proportionate response of $NO_y$-N dry deposition to $NO_x$ emissions variations[17,48]. Our study, however, suggests that over the recent decade, the decreases of $NO_x$ concentrations in the atmosphere cannot be directly linked to changes in $NO_y$-N dry deposition on a national scale, because the reduction of $NO_y$-N deposition under the $NO_x$ emission control scenarios could be unexpectedly low for Eastern China though efficient for Southern China. The weakened response of $NO_y$-N deposition to $NO_x$ emission reductions should be taken into account when assessing the spatiotemporal patterns of total reactive N deposition fluxes and the ecosystem responses to the regulation of $NO_x$ emissions.

## Methods
**Chemical-transport model experiments**. The regional atmospheric chemical-transport model, WRF-Chem (version 3.6.1), was used in this study to reproduce the emissions, transport, photochemical reactions, and deposition of the reactive nitrogen species[49]. The simulation domain covered the whole Chinese terrestrial land and surrounding oceanic regions (the small islands in the South China sea are not included in this study) with a horizontal resolution of 50 km × 50 km and 24 vertical layers from the surface to 50 hPa. The simulations were conducted for the whole year of 2015 and each model run covered 84 h with a 12-h spin-up time. The initial and boundary meteorological conditions were provided by 6-h National Centers for Environmental Prediction reanalysis data with 1° × 1° spatial resolution. We used the CBMZ mechanism to simulate gas-phase photochemical reactions[50] and the MOSAIC module with four discrete aerosol size bins (0.039–0.156, 0.156–0.624, 0.624–2.5, and 2.5–10.0 μm in dry diameter) to simulate aerosol microphysical and chemical processes[51]. The combination of CBMZ and MOSAIC mechanisms has been demonstrated to show a good performance in simulating nitrate concentrations over China in our previous studies[10,52]. Biogenic emissions were estimated online by WRF-Chem using the Model of Emissions of Gases and Aerosols from Nature[53]. The dry and wet deposition processes of tracers and aerosols including in-cloud and below-cloud wet removal have been treated in the standard version of WRF-Chem[54]. We calculated in our simulations that the region-averaged dry deposition velocities of $NO_2$ and gaseous nitric acids were 0.12 and 1.5 cm s$^{-1}$, respectively, in Eastern China, and 0.12 and 2.3 cm s$^{-1}$ in Southern China. These estimates were broadly consistent with existed observations and three-dimensional model simulations in similar terrestrial ecosystems[55,56]. The detailed model configuration used in this study can be found in ref. [57] and ref. [10].

To investigate the response of reactive N deposition to the $NO_x$ emission reductions, we performed several parallel simulation experiments with different $NO_x$ emission scenarios for China (Supplementary Table 1). For the Baseline simulation, the anthropogenic emissions of the year 2015 were taken from the

MEIC emission database[9], in which the national $NO_x$ emission amount was 23.7 Tg yr$^{-1}$, about 18% lower than the emission (29.0 Tg yr$^{-1}$) in 2011. The database shows that China's $NO_x$ emissions peaked in 2012 and then decreased by about 40% until 2019. The Chinese Statistics Yearbook (http://www.stats.gov.cn/tjsj/ndsj/) shows that $NO_x$ emissions in China has decreased by 46% from 2012 to 2017. In the $NO_x$ reduction cases, we specified the changes in $NO_x$ emissions of −10, −30, and −50% relative to the emissions in 2015. These simulation experiments were made by reducing the $NO_x$ emissions in each grid cell by 10, 30, and 50% for the three scenarios (referred to as RED10, RED30, and RED50), respectively. As emissions can be transported away from source regions (in this study mainly including Eastern China, Southern China, and the other domestic region) and deposited to receptor regions, the relative contributions of different emission sources to the deposition at the targeted regions were calculated. We separately switched off the emissions at each source region in the parallel simulations and compared the results between them and the Baseline case (Supplementary Table 1).

Besides, to indicate the trend in $NO_y$-N deposition given both $NO_x$ and volatile organic compounds emission variation in the future, we employed the central emission pathway (SSP245-ECP for 2050) developed by ref. [58] that have fully considered the future socio-economic and climate developments and local pollution control policies. For this pathway, the $NO_x$ and VOCs emissions in China are projected to be reduced by 48 and 34% from 2015 to 2050.

**Observation datasets**. The Nationwide Nitrogen Deposition Monitoring Network (NNDMN) at 24 stations across China was used in this study to demonstrate the characteristics of reactive N deposition and evaluate the model performance. The dataset provided the monthly accumulated bulk (wet) deposition of gas and particle $NO_3$-N and $NH_x$-N and surface-air concentrations of major N-containing compounds from the year 2011 to 2015. The dry N deposition fluxes in this database were calculated offline using the dry deposition velocities of gaseous nitric acid and particulate nitrate given by a chemical-transport model. The full description of the sampling methods and associated sampling errors are shown in refs. [23,55]. Besides, the daily measurements of atmospheric nitrate concentrations in particulate matter with a dry diameter of less than 2.5 μm were collected at 35 sites covering Eastern China, Southern China, and Southwestern China to validate the modeled nitrate concentrations. The comparison of modeled nitrate mass concentrations with this dataset was provided in this study. Description of the site information can be seen in our recent study[10]. We also used the QA4ECV monthly-averaged tropospheric $NO_2$ columns measured by ozone monitoring instrument onboard NASA Aura (available online at: http://www.temis.nl) to reflect the inter-annual trend in $NO_2$ concentrations over China during 2011-2015[59].

**The Community Land Model**. In this study, we quantified the effects of N deposition fluxes reduction on terrestrial carbon and nitrogen dynamics over China using a widely used model, the Community Land Model version 5 (CLM5)[60,61]. The CLM simulates land biogeophysical, biogeochemical, and landscape processes, including surface energy, water, carbon, nitrogen, momentum, and radiative fluxes over different land types and ecosystems[62]. Vegetated surfaces are comprised of up to 15 possible natural plant functional types, 64 crop functional types plus bare ground. Based on the Land Use Harmonized version 2 transient datasets, we considered the changes in the area fractions of different plant functional types and crop functional types during the historical period from 1850 to 2014. Photosynthesis was simulated by the Farquhar model[63], which can represent an increase in the photosynthetic rate under elevated $CO_2$ concentrations. Plant growth depends on the carbon uptake through photosynthesis as well as nitrogen availability from soils. Rising nitrogen deposition potentially increases the soil nitrogen availability and therefore releases the nitrogen limitation for plant growth. In comparison to previous CLM versions, the CLM5 uses the Fixation and Uptake of Nitrogen module, which considers the carbon cost in nitrogen acquisition. The improved vegetation nitrogen-carbon interactions and vegetation-soil nutrient competition in CLM5 enable us to pinpoint the response of ecosystem carbon balance to varying N deposition inputs to the land[64]. The performance of the CLM5 carbon cycle has been systematically evaluated in the International Land Model Benchmarking projects on both site scale and global scale[60,65].

The model simulation was initialized at a spatial resolution of 2.5° longitude by 1.875° latitude in 1850 through a spin-up, which recursively use reanalysis meteorological forcing, the CRUNCEP version 7 set from 1901 to 1920, to drive ecosystem carbon towards the steady-state. Then, historical simulations were conducted from 1850 to 2014 with rising atmospheric $CO_2$ concentration, warming air temperature, land-use change, and increasing nitrogen deposition. Plant hydraulics, crop, and fire modules were turned on in conjunction with the nitrogen modules. To identify the effects of decreased N deposition on the terrestrial ecosystem, we conducted three scenarios of transient simulation experiments in the period between 2005 and 2014 with different N deposition from WRF-Chem as inputs: (1) N deposition in the Baseline case; (2) N deposition in the RED30 case, in which $NO_x$ emissions over China were reduced by 30% but N deposition changes smaller (by about 20%); (3) N deposition reduced by 30% nationwide relative the Baseline case, corresponding to a 100% relative response of N deposition to $NO_x$ emission reductions. We aim to estimate the natural carbon sink in response to the reductions of N deposition over China by human activities and to show implications for China's future carbon budgets and solutions to carbon neutrality.

## Data availability

The Nationwide Nitrogen Deposition Monitoring Network data are publicly available at https://doi.org/10.1038/s41597-019-0061-2. Modeled nitrogen deposition results are available in an open-access repository (https://doi.org/10.5281/zenodo.4727591).

## Code availability

The WRF-Chem source codes are freely available for download at https://ruc.noaa.gov/wrf/wrf-chem/. Figures were created mainly based on NCAR Command Language (NCL, https://www.ncl.ucar.edu/) and the codes are available from the corresponding authors upon request.

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

## Acknowledgements

We thank Prof. A. R. Ravishankara (Departments of Chemistry and Atmospheric Science, Colorado State University) for valuable suggestions and improvement on the manuscript. This study was funded by the National Natural Science Foundation of China (NSFC) (91644212 and 42075180), the National Key R&D Program of China (2016YFC0201505 and 2017YFC0210103), and National Research Program for Key Issues in Air Pollution Control (DQGG0208). We acknowledge Dr. Yongjie Huang (IAP/CAS) for providing the map database.

## Author contributions

Y.S. and T.Z. conceived and led the study. M.L., F.S., X.Lu., and Y.S. performed research and wrote the draft. M.L., F.S., X.H., B.L., Q.Z., X.Liu, J.C., T.X., T.W., Z.X., W.X., W.L., L.K., X.C., and H.Z. analyzed data and interpreted the results. X.Lu and Y.D. conducted the CLM model simulations and analyzed the data. All authors commented on the manuscript.

## Competing interests

The authors declare no competing interests.
