## [Peer Review File · Nature Communications]

Unexpected response of nitrogen deposition to nitrogen oxide controls and implications for land carbon sinkReviewers' comments:

Reviewer #1 (Remarks to the Author):

This study used a regional chemical transport model with a nationwide N deposition observation network to simulate the response efficiency of atmospheric N deposition to NO_x emission controlling policy in China. They found weakened responses of N deposition to NO_x emission reduction in China, especially in Eastern China. This topic is interesting and necessary. However,

I still have some major concerns listed as follow.

The main innovative results of the manuscript are a little bit outdated. Some recent studies have reported the lag response of N deposition to N emission reduction both in some regions and whole China. For example, Fu et al., 2020, Zhong et al., 2020, Wen et al., 2020, Xi et al., 2021. Therefore, I would hesitate to recommend this manuscript to be accepted in Nature Communication.

Fu et al., 2020. Persistent Heavy Winter Nitrate Pollution Driven by Increased Photochemical Oxidants in Northern China. *Environ. Sci. Technol.* 54, 3881-3889.

Zhong et al., 2020. Meteorological variations impeded the benefits of recent NO_x mitigation in reducing atmospheric nitrate deposition in the Pearl River Delta region, Southeast China. *Environ. Pollut.* 266, 115076.

Wen et al., 2020. Changes of nitrogen deposition in China from 1980 to 2018. *Environ. Int.* 144, 106022.

Xi et al., 2021. Hysteresis response of wet nitrate deposition to emission reduction in Chinese terrestrial ecosystems. *Atmos. Environ.* [https://authors.elsevier.com/sd/article/S1352-2310\(21\)00377-0](https://authors.elsevier.com/sd/article/S1352-2310(21)00377-0).

I suggest the authors to rethink the significance of this study and move forward with new simulation scenarios, i.e. changing meteorological condition with fixed emission, changing emission with fixed meteorological condition. The authors mentioned that this study only considered the effects of anthropogenic emissions on reactive N deposition, while the effect of climatological conditions, such as variability of precipitation, did not be considered. We already knew that the decreased of NO_x emission driven by the strict environment policy in China did not decreased wet nitrate deposition or bulk N deposition as expected. An increase in O₃ concentrations during 2013-2017 in China has also been reported. The enhancement of the atmospheric oxidation capacity accelerates the conversion of NO_x to HNO₃. In addition, the reduction of SO₂ will reduce the consumption of OH converted to sulfate, which may indirectly increase the oxidation capacity of the atmosphere. High level of NH₃ emissions in China provide sufficient precursors for the further conversion of HNO₃ to nitrate. The combined pollution is more complicated than we thought. We are eager to know what the roles of climatological conditions play in N deposition under different NO_x emission reduction scenarios? What is the threshold value of emission reduction level that the air quality will stay high even with bad climatological conditions?

Line 97: Confused. Please clarify. The classification of N deposition in Table S1 is hard to follow. In general, atmospheric deposition includes wet and dry deposition based on deposition way. The form of N in the atmosphere includes gaseous NH₃, gaseous NO₂, gaseous HNO₃, particulate NH₄⁺, particulate NO₃⁻, and organic N. More detail information need to be added in this table.

Line 121-125: The response efficiency of NO₃-N dry deposition to NO_x emission was 62–85% on a national scale, while the decreases of NO₃-N wet deposition were almost proportionate to NO_x emission reductions with the corresponding effectiveness of 92–99%. This result was inconsistent with Wen et al. 2020, which found that dry oxidized N deposition decreased significantly from 2011-2018 due to NO_x emission controls. The model results did not capture the observation results?

Line 332: Maybe it is a good choice to show the detail information of parallel simulation experiments as a Table.

Line 339: “from 2015 to present”, what do you mean present here?

Reviewer #2 (Remarks to the Author):

This paper addresses how reductions in reactive nitrogen emissions in China under different mitigation scenarios may have varying effectiveness and impacts on reducing nitrogen deposition. The conclusion is definitely significant and novel, and has major implications on the evaluation of future emission strategies to minimize impacts on terrestrial ecosystems. There are, however, some important questions that need to be addressed before this paper can be published. See below.

1. The underlying mechanisms examined here behind the varying responses of nitrogen deposition on NO_x reduction in Eastern vs. Southern China have indeed been well researched and documented in previous work, as the authors have also cited – namely, the enhanced oxidant levels due to reduced NO_x in Eastern China in the VOC-limited regime (i.e., reduced “titration” effect). I understand that previous works focused on the enhanced ozone levels following NO_x reductions, and this paper extended this further to examine the subsequent impacts on HNO₃ formation and deposition, but this new result seems incremental compared to existing literature. To be publishable in Nature Communications, more justification is needed to explain how the new result regarding nitrogen deposition, derived from known mechanisms, would substantially revise previous understanding and have new implications on policy making. One possible way, among others, to do so would be to articulate how ecosystem impact assessment done before would be affected by the new result would, and how such assessment should

be done in the future. These have been superficially mentioned, but a lot more references and discussion of terrestrial ecosystem-atmosphere nitrogen interactions are warranted (see below also).

2. One novel aspect and important implication of this paper is the potential impacts of future emission control strategies on terrestrial ecosystems (e.g., vegetation productivity, soil biogeochemistry, soil acidification, etc.). Thus far, these are mentioned only peripherally, but it should stand at the center of the main motivation/impact of the paper. At least a paragraph or two in the introduction and discussion are warranted to discuss how the varying responses of nitrogen deposition in Eastern vs. Southern China may affect different ecosystem processes (e.g., primary production, carbon uptake, soil nitrification and denitrification, soil pH, etc.), and suggest possible emission reduction strategies that can maximize benefits for both human health and ecosystem health.

3. As nitrogen deposition influences ecosystems tremendously, while vegetation and soil in ecosystems play major roles regulating surface air quality and atmospheric composition, there can be various feedback mechanisms whereby NO_x reductions will lead to changes in nitrogen deposition, thus vegetation and soil, that ultimately affect air quality itself. Such feedback mechanisms were suggested by Zhao et al. (Zhao, Y. H., Zhang, L., Tai, A. P. K., Chen, Y. F., and Pan, Y. P.: Responses of surface ozone air quality to anthropogenic nitrogen deposition in the Northern Hemisphere, *Atmos. Chem. Phys.*, 17, 9781-9796, doi:10.5194/acp-17-9781-2017, 2017), and can be potentially important. However, the current modeling framework in this paper would not allow such feedbacks to be examined because the land cover is mostly prescribed. At least the implications of these feedback mechanisms on the main results of this paper should be discussed.

4. The authors used one model to derive the results. However, multi-model differences are bound to occur, which may or may not annihilate the key conclusions of this paper. It is understandable that the results regarding nitrogen deposition per se are novel and thus hard to compare with previous results since there were none, but the authors should cross-compare their simulated results in other variables that would lead directly to the results regarding nitrogen deposition, e.g., ozone concentrations, HO_x concentrations, etc., with previous works (many of which were cited, but not compared). This would give greater credence to the validity of the new results.

Reviewer #3 (Remarks to the Author):

General.

This study analyses the mechanism of a non-linear response of reactive nitrogen deposition (in particular in Eastern China) to NO_x emission reductions. While the issue of non-linear responses is well known in the atmospheric chemistry community, the interesting aspect is that emission changes are so large in a relatively short time (e.g. a downward emission trend of 20 % or more), making these feedbacks more visible than would be the case under gradually changing conditions.

The manuscript is interesting, but there are several loose ends that require to be corroborated, additional testing and analysis.

Specifically:

1) Given spatial and temporal variability, the observational evidence of using only 4 deposition stations over Eastern China for 5 years is very limited. If more data is available also for earlier years, they need to be analysed as well to better understand the trends and variability from 2010-2015. One possible model simulation to demonstrate variability is keeping emissions constant, and analyse the model responses.

2) There is an interesting discrepancy between NO₂ column observations and the anthropogenic emissions trends (e.g. Figure 1). This needs to be explored further along with an calculation on how representative NO₂ is for NO_y dry deposition, and how it is changing. This is relevant for emerging studies that use satellite observations to compute deposition of NO_y (and NH_x).

3) Why hasn't NH₃ satellite column observations not been used for analysis (given the probably important role in buffering some of the excess nitrate produced).

4) The role of atmospheric transport in feedback on deposition between regions has not been discussed. Perturbations similar to MICS-Asia, HTAP etc, where only emissions over East China, South China and the rest of China, would be helpful to explore transport feedbacks

5) Appropriate budget analysis possibly combined with point 4, would be helpful to understand the closure of emissions, chemistry, transport and removal. An analysis of overall NO_y lifetime (for China, and its subregions) would be helpful as well.

6) In view of point 5, an appropriate analysis of the chemical budget terms (in particular all the loss reactions NO₂+OH; NO₂+O₃; N₂O₅ heterogenous) would be useful. The papers main hypothesis is that due to a shift of NO to NO₂ (NO_x component), and also more OH and other oxidants, the loss of NO_y is speeding up. This can only work if subsequently the HNO₃ is not forming aerosol nitrate. The paper is ignoring an analysis of this and the analysis of the nitrate (HNO₃, and NH₄NO₃ and other nitrates) must be shown.

7) At several places in the publication the word effectiveness needs to be replaced, possibly by relative response or similar.

I do think the work can be of interest to Nature if my major comments above, and detailed comments below are duly taking into account.

Detailed comments

I. 34 clarify largest in budget terms- or in terms of deposition fluxes?

I. 37 It is not clear what is meant with 'integrating a model with observation'- I think the authors mostly compare (with the exception of dry dep, that is using modelled deposition velocities. Please clarify.

I. 39 what was expected on the basis of what? The word expect suggests that an initial evaluation was made. I suggest 'unforeseen' as more appropriate. In comparing USA, Southern and Eastern China, it should be discussed that they have a very different geographical extent and make up of emissions within the regions, which makes such numbers somewhat comparing apples and pears.

I. 43-44 should be considered. I think most if not all models would include such feedbacks through atmospheric chemistry, but I leave it open whether these models have the correct response due to issues with model resolution, transport, chemistry, emissions. It is likely that the model study presented here, was more accurate than e.g. used in MICS Asia, HTAP exercises, but that remains to be demonstrated.

I. 54 I recommend to use consistently in this study the units NO_x-N and NH₃-N for emissions. Currently it is not clear what this, most like NO_x expressed as NO₂. If the authors want to use these units, the unit need to be 30 Tg Nox(NO₂). yr⁻¹.

I. 57 Indeed the Qian paper report stagnant emissions for NH₃, but I recommend to be somewhat careful with this, as that paper doesn't provide much information on what basis this assertion was made. A rough cross check could be made by comparing to trends in fertilizer production and imports and legumes import as a proxy for activity data that are ultimately causing NH₃ emissions (in the absence of changing management and controls). As this could possibly change the outcomes of the study, it would be good to pay more attention to this aspect.

L 69 nitric acid and (neutralized) aerosol nitrate and organic nitrates. List is not complete

I. 70 OH react with NO₂ (not with NO_x in general). O₃ reacts with NO to form NO₂. O₃ reacts with NO₂ to form NO₃, which under light conditions is rapidly photolysed. Please be more exact.

I. 89 See comment on figure S1.

I. 92 social-economic=>just economic is sufficient in this context

I. 96 so NH₃-N deposition 5.8 Tg N? Please mention this explicitly. Comparing to the earlier mentioned NO_x-N emission of 9.13 and NH₃-N emissions of 8.2 Tg N, a back of the envelop calculation gives 57 % of NO_x deposited on land- and 63.5 % of ammonia N? A budget analysis (including changes could be developed as strong key message of this paper) as the fraction remaining on land.

I. 97 here the authors should be more specific on the form of NO₃: as gaseous HNO₃ or in the form of aerosol nitrate? The specific form influences deposition strongly and we need to know whether this is part of the response function.

I. 103 see comments to Figure S2

I. 107 a substantial change of NO_y lifetime will depend on the ability to form aerosol ammonium (and other) nitrates. It is not clear what is compared in S2 (gas, aerosol, both?) and I think it is not nitrite/nitrous acid as erroneously in S2.

I. 109. Which year is 'now'?

I. 112 what is the baseline case? Emissions constant or fluctuating over the years? Clarify shortly here, and methods for more details.

I. 116 see comment to Figure 2. The paper really needs to explore better the role of change in residence time and the role of atmospheric transport, including monsoon patterns.

I. 119 I think effectiveness is not the correct word- response ratio is more appropriate.

I. 122 this would point to more NO₃ in the form of aerosol nitrate, which would work to prolong lifetime?

I. 143 what is present?

I. 166 A thorough chemical budget analysis would pinpoint these hypothesis.

I. 333 this is the response to the earlier question (single year of emission or variable), please summarize in earlier sentence.

I. 357 what is measured. Aerosol nitrate, gaseous nitric acid or both together. Duly discuss possible sampling errors.

I. 365 Why was not a similar comparison with NH₃ made?

Fig 1: Please define coordinates of 'Eastern China' or refer to Figure 2. It would also be interesting to present the domain integrated deposition fluxes, possibly in the same units Tg NO_y-N, and NH₃-N. It is not clear why only 4 stations were selected.

Fig 2: Is this a 4 year average? Does panel a) include Eastern China. It would probably be more instructive to have a separate panel for 'other China'. What exactly is displayed in 2c? I think it is a scaled response to 30 % emission reduction, not an effectiveness, as part of the response can be due to transport. Also it took me until figure 3 to understand that the green colors over Eastern China are actually higher deposition. Probably it makes sense to first Figure 3 and then 2? Or add a sentence in the figure caption guiding the readers' interpretation.

Fig 3: the enhanced response of NO₃ deposition over Eastern China, in particular in winter, raises question what the chemical mechanism really is. Conventional wisdom points to heterogenous (N₂O₅

possibly also NO₃ radical) reactions as the dominant driver of NO_x removal, but it is not clear why these reaction rates would be enhanced in winter. Unfortunately the paper doesn't do a thorough job in analyzing these chemical mechanism when attempting to understand the drivers of these responses, although there may be clues in Figure 4, where in particular NO₃ radical seems to have a strong response. More analysis needed.

Figure 4: You should also plot (numerically) the absolute values of O₃, NO₃ and OH, to get a feeling of the importance of the numbers.

Figure 4: What is going on over Tibet with ratios close to zero "Black stars in figure 2c refer to the four observations in Figure 1?

Fig S1 These are nice plots, but it is difficult to understand the model skills. I recommend to include 2 additional panels that provide a scatter plot+some spatial skill statistics. Also to exclude variability issues it would be better to evaluate skills over a 5 year period rather than a single year.

Figure S2 Probably the authors meant nitrate (not nitrite). The way this is presented suggest better model performance. A commonly used statistic is also the fraction of model results with a factor of 0.5 and 2 from observations.

Response to Reviewer #1

Reviewer #1 (Remarks to the Author):

Reviewer #1: This study used a regional chemical transport model with a nationwide N deposition observation network to simulate the response efficiency of atmospheric N deposition to NO_x emission controlling policy in China. They found weakened responses of N deposition to NO_x emission reduction in China, especially in Eastern China. This topic is interesting and necessary. However, I still have some major concerns listed as follow.

Response: We appreciate the reviewer's comments on our manuscript. Please see our point-to-point responses in the following.

Reviewer #1: The main innovative results of the manuscript are a little bit outdated. Some recent studies have reported the lag response of N deposition to N emission reduction both in some regions and whole China. For example, Fu et al., 2020, Zhong et al., 2020, Wen et al., 2020, Xi et al., 2021. Therefore, I would hesitate to recommend this manuscript to be accepted in Nature Communication.

Response: The novelty of this study includes:

- 1) Our manuscript demonstrated that the NO_x controls enhanced atmospheric oxidation capacity and subsequently increased oxidized N deposition in China. The weakened response of atmospheric N deposition to the rapid NO_x reductions and the chemical mechanisms embedded in have been NOT reported. All these previous studies mentioned by the reviewer did not give the answers. Fu et al. (2020) even did not mention N deposition. Please see the detailed analysis in the Line 184-250 in the revised manuscript.
- 2) In the revised manuscript, we performed the land model simulations using CLM5 and gave associated analysis on the regional ecosystem responses to the changes in N deposition. These new results advance our understanding on the future trends in China's land carbon sinks and the pathways to achieve carbon neutrality goal. Please see Line 298-331 and Line 345-359 in the revised manuscript.

Reviewer #1: I suggest the authors to rethink the significance of this study and move forward with new simulation scenarios, i.e. changing meteorological condition with fixed emission, changing emission with fixed meteorological condition. The authors mentioned that this study only considered the effects of anthropogenic emissions on reactive N deposition, while the effect of climatological conditions, such as variability of precipitation, did not be considered. We already knew that the decreased of NO_x emission driven by the strict environment policy in China did not decreased wet nitrate deposition or bulk N deposition as expected. An increase in O₃ concentrations during 2013-2017 in China has also been reported. The enhancement of the atmospheric oxidation capacity accelerates the conversion of NO_x to HNO₃. In addition, the reduction of SO₂ will reduce the consumption of OH converted to sulfate, which may indirectly increase the oxidation capacity of the atmosphere. High level of NH₃ emissions in China provide sufficient precursors for the further conversion of HNO₃ to nitrate. The combined pollution is more complicated than we thought. We are eager to know what the roles of climatological conditions play in N deposition under different

NO_x emission reduction scenarios? What is the threshold value of emission reduction level that the air quality will stay high even with bad climatological conditions?

Response: This study shows that enhanced oxidation capacity caused by NO_x controls would increase the NO_y deposition rates. Those several factors proposed by the reviewer do NOT affect our conclusions. Please see the following points.

- 1) We perform five-year simulations with the constant emissions at 2015 but varying meteorological conditions from 2011–2015. The model results demonstrate that meteorological factors gave rise to diverse temporal variabilities in N deposition at those observation stations. Only meteorological condition variations cannot explain the weakened response of NO_y deposition to NO_x emissions. This study proposes that the enhanced atmospheric oxidation capacity is a key driving factor. Please see Line 124-135.
- 2) The major conclusions are unchanged even we use different climatological conditions (e.g., precipitation and monsoon). The sensitivity simulations using the meteorological inputs for 2011, in which monsoon circulation was likely different with that in 2015 (Li et al., 2016; Zhang et al., 2019), show that the percent changes of NO_y deposition with the 30% NO_x reduction was 17% in the Eastern China and 26% in the Southern China, in line with the simulations for 2015. Please see Line 161-167.
- 3) We have shown that both the decreased NO_x and SO₂ emissions would increase the atmospheric oxidation capacities in our recent studies (Please see Liu et al., 2018, 2019, and 2020). The decreases of SO₂ emissions and associated sulfate formation via the OH• + SO₂ pathway can increase particulate nitrate fractions in the total nitrate with increased ammonia availability (Please see Liu et al., 2018 and Zhai et al., 2021); however, this would decrease NO_y deposition because the deposition rate of aerosol nitrate is much lower than gas-phase nitrate. This study reveals that it was the NO_x emission reduction that enhanced oxidation capacities and consequently increased the NO_y dry deposition rate.
- 4) Though the emission control strategies to improve air quality under different climatological conditions was not the topic of this study, our results from the view of N deposition indicate that the coordinated controls for NO_x and VOCs could reduce nitrate concentrations effectively despite the unfavorable meteorological conditions. Please see Line 283-297.

Reviewer #1: Line 97: Confused. Please clarify. The classification of N deposition in Table S1 is hard to follow. In general, atmospheric deposition includes wet and dry deposition based on deposition way. The form of N in the atmosphere includes gaseous NH₃, gaseous NO₂, gaseous HNO₃, particulate NH₄⁺, particulate NO₃⁻, and organic N. More detail information need to be added in this table.

Response: Accepted. We reword this sentence. More detailed information is added in the table. Please see Line 109-110 and Table S3 in the revised manuscript.

Reviewer #1: Line 121-125: The response efficiency of NO₃-N dry deposition to NO_x emission was 62–85% on a national scale, while the decreases of NO₃-N wet deposition were almost proportionate to NO_x emission reductions with the corresponding effectiveness of 92–99%. This result was inconsistent with Wen et al. 2020, which found that dry oxidized N deposition decreased

significantly from 2011-2018 due to NO_x emission controls. The model results did not capture the observation results?

Response: Our model did capture the observations. Even though the response efficiency of NO₃-N dry deposition was as low as 62%, the rapid NO_x reduction led to an apparent decrease (−21%) of dry oxidized N deposition on a nation scale, consistent with the observation (−25%) in Wen et al., 2020.

Reviewer #1: Line 332: Maybe it is a good choice to show the detail information of parallel simulation experiments as a Table.

Response: Accepted. We describe all simulation experiments within a table in Supplementary materials. Please see the detailed description in Table S1.

Reviewer #1: Line 339: “from 2015 to present”, what do you mean present here?

Response: Accepted. We reword this sentence. Please see Line 440-441 in the revised manuscript.

References:

- Fu et al., 2020. Persistent Heavy Winter Nitrate Pollution Driven by Increased Photochemical Oxidants in Northern China. *Environ. Sci. Technol.* 54, 3881-3889.
- Li Q, Zhang R, Wang Y. Interannual variation of the wintertime fog-haze days across central and eastern China and its relation with East Asian winter monsoon. *Int J Climatol* 36, 346-354 (2016).
- Liu M, et al. Rapid SO₂ emission reductions significantly increase tropospheric ammonia concentrations over the North China Plain. *Atmos Chem Phys* 18, 17933-17943 (2018).
- Liu M, et al. Ammonia emission control in China would mitigate haze pollution and nitrogen deposition, but worsen acid rain. *Proceedings of the National Academy of Sciences* 116, 7760 (2019).
- Liu M, et al. Trends of Precipitation Acidification and Determining Factors in China During 2006–2015. *J Geophys Res-Atmos* 125, e2019JD031301 (2020).
- Xi et al., 2021. Hysteresis response of wet nitrate deposition to emission reduction in Chinese terrestrial ecosystems. *Atmos. Environ.* [https://authors.elsevier.com/sd/article/S1352-2310\(21\)00377-0](https://authors.elsevier.com/sd/article/S1352-2310(21)00377-0).
- Wen et al., 2020. Changes of nitrogen deposition in China from 1980 to 2018. *Environ. Int.* 144, 106022.
- Zhang G, et al. Seesaw haze pollution in North China modulated by the sub-seasonal variability of atmospheric circulation. *Atmos Chem Phys* 19, 565-576 (2019).
- Zhong et al., 2020. Meteorological variations impeded the benefits of recent NO_x mitigation in reducing atmospheric nitrate deposition in the Pearl River Delta region, Southeast China. *Environ. Pollut.* 266, 115076.

Response to Reviewer #2

Reviewer #2 (Remarks to the Author):

This paper addresses how reductions in reactive nitrogen emissions in China under different mitigation scenarios may have varying effectiveness and impacts on reducing nitrogen deposition. The conclusion is definitely significant and novel, and has major implications on the evaluation of future emission strategies to minimize impacts on terrestrial ecosystems. There are, however, some important questions that need to be addressed before this paper can be published. See below.

Response: We appreciate the reviewer's useful comments. All the comments are accepted and fully addressed. Please see our revisions in the following.

Reviewer #2: 1. The underlying mechanisms examined here behind the varying responses of nitrogen deposition on NO_x reduction in Eastern vs. Southern China have indeed been well researched and documented in previous work, as the authors have also cited – namely, the enhanced oxidant levels due to reduced NO_x in Eastern China in the VOC-limited regime (i.e., reduced “titration” effect). I understand that previous works focused on the enhanced ozone levels following NO_x reductions, and this paper extended this further to examine the subsequent impacts on HNO₃ formation and deposition, but this new result seems incremental compared to existing literature. To be publishable in Nature Communications, more justification is needed to explain how the new result regarding nitrogen deposition, derived from known mechanisms, would substantially revise previous understanding and have new implications on policy making. One possible way, among others, to do so would be to articulate how ecosystem impact assessment done before would be affected by the new result would, and how such assessment should be done in the future. These have been superficially mentioned, but a lot more references and discussion of terrestrial ecosystem-atmosphere nitrogen interactions are warranted (see below also).

Response: Accepted. We add detailed and quantitative assessment on how the reduced N deposition impacts terrestrial ecosystems, and show new implications on policy making, especially for China's carbon neutral goal.

- 1) We perform the land model simulations using the Community Land Model version 5 (CLM5) with the inputs of N deposition from WRF-Chem simulations for different cases. The plant N uptake, net primary production (NPP), and net ecosystem production (NEP; used as an indicator of land carbon sink) from CLM5 were analyzed to evaluate the effects of reduced N deposition on terrestrial ecosystems.
- 2) The new results based on CLM5 demonstrated that the 30% reduction of NO_x emission over China decreases N deposition by 1.4 Tg N yr⁻¹ and subsequently decrease the NEP by 5.0 Tg C yr⁻¹ through the limitation of plant N uptake and plant growth. We find that the unexpected weakened response of N deposition would lead to much less carbon sequestration in land ecosystems in the Eastern China compared the ideal case with a 100% relative response of N deposition (–0.56 Tg C yr⁻¹ vs. –0.04 Tg C yr⁻¹).
- 3) Our new analysis indicates that the projected emission reductions for NO_x to improve air quality in China diminish ecosystem production. This N-control penalty may represent 5–10% of required land carbon sink to achieve carbon neutral over China by the next mid-century (Wang

et al., 2014; Cheng et al., 2020). To our knowledge, it is the first time to project the impact of future controls of N emissions on natural carbon sink.

- 4) As suggested by the reviewer's next comments, the N deposition-ecosystem interactions are discussed in details (with references like Zhao et al., 2017; Sadiq et al., 2017; Zhou et al., 2018). Please see Line 360-374 in the revised manuscript.

All these revisions have been updated in the revised text. Please see Line 298-331, Line 345-387, and Line 465-487 in the revised manuscript.

Reviewer #2: 2. One novel aspect and important implication of this paper is the potential impacts of future emission control strategies on terrestrial ecosystems (e.g., vegetation productivity, soil biogeochemistry, soil acidification, etc.). Thus far, these are mentioned only peripherally, but it should stand at the center of the main motivation/impact of the paper. At least a paragraph or two in the introduction and discussion are warranted to discuss how the varying responses of nitrogen deposition in Eastern vs. Southern China may affect different ecosystem processes (e.g., primary production, carbon uptake, soil nitrification and denitrification, soil pH, etc.), and suggest possible emission reduction strategies that can maximize benefits for both human health and ecosystem health.

Response: Accepted. We add substantial results and discussions on how future emission controls would impact different ecosystem processes. Please see the major revisions below and more details in the revised manuscript:

- 1) As aforementioned, we assess the effects of NO_x emission controls on terrestrial production and carbon uptake by coupling the ecosystem model (CLM5) with the simulated N deposition.
- 2) The results based on CLM5 show that the decrease of NEP induced by the N deposition reduction in the Eastern China (0.73 g C m⁻² yr⁻¹) is appreciably lower than that in the Southern China (1.4 g C m⁻² yr⁻¹), reflecting the divergence in their biome types and associated plant growth. On the other hand, the future projected reductions in anthropogenic emissions of both reduced and oxidized nitrogen in China would mitigate the risks of excess N inputs on some ecosystems, particularly for the Southern China (Liu et al., 2019).
- 3) Our findings along with some previous studies highlight the coordinated controls of NO_x and volatile organic compounds to reduce atmospheric oxidation capacity (e.g., ozone pollution) and its harmful effects on human health and ecosystem production; importantly, the new implication is that the N-control penalty on natural carbon sink via plant N uptake should be considered in the pathways towards carbon neutrality, as it is possibly on the order of 5–10% of China's land carbon sink.

Please see detailed revisions at Line 298-331 in the revised manuscript.

Reviewer #2: 3. As nitrogen deposition influences ecosystems tremendously, while vegetation and soil in ecosystems play major roles regulating surface air quality and atmospheric composition, there can be various feedback mechanisms whereby NO_x reductions will lead to changes in nitrogen deposition, thus vegetation and soil, that ultimately affect air quality itself. Such feedback mechanisms were suggested by Zhao et al. (Zhao, Y. H., Zhang, L., Tai, A. P. K., Chen, Y. F., and Pan, Y. P.: Responses of surface ozone air quality to anthropogenic nitrogen deposition in the Northern Hemisphere, Atmos. Chem. Phys., 17, 9781-9796, doi:10.5194/acp-17-9781-2017, 2017),

and can be potentially important. However, the current modeling framework in this paper would not allow such feedbacks to be examined because the land cover is mostly prescribed. At least the implications of these feedback mechanisms on the main results of this paper should be discussed.

Response: Accepted. We add one paragraph discussing N deposition-ozone feedbacks mediated by NO_x emission controls, as mentioned by Zhao et al. (2017). Considering the complex biosphere-atmosphere interactions, we summarize that NO_x emission controls can affect air quality (mainly ozone pollution) through the direct effects of NO_x-VOCs chemistry on ozone formation and indirectly through the N deposition effects on plant growth (changing biogenic VOCs emissions and ozone dry deposition velocities) and soil NO_x emissions (Zhao et al., 2017; Sadiq et al., 2017; and Zhou et al., 2018). Please see Line 360-374 in the revised manuscript.

Reviewer #2: 4. The authors used one model to derive the results. However, multi-model differences are bound to occur, which may or may not annihilate the key conclusions of this paper. It is understandable that the results regarding nitrogen deposition per se are novel and thus hard to compare with previous results since there were none, but the authors should cross-compare their simulated results in other variables that would lead directly to the results regarding nitrogen deposition, e.g., ozone concentrations, HO_x concentrations, etc., with previous works (many of which were cited, but not compared). This would give greater credence to the validity of the new results.

Response: Accepted. Our simulation results for the Baseline case are validated by many observation datasets, including nationwide N deposition networks and in-situ atmospheric measurements of trace gases and aerosols near the ground. The model performance of N deposition in the Baseline case was also close to the ensemble mean of multi-model results in the model comparison study MICS-Asia III (Ge et al., 2020). Please see Figure S1-S2, Table S2, and Line 90-108 in the revised manuscript.

References:

- Cheng J, et al. Pathways of China's PM_{2.5} air quality 2015–2060 in the context of carbon neutrality. *Natl Sci Rev*, (2021).
- Ge et al., Model Inter-Comparison Study for Asia (MICS-Asia) phase III: multimodel comparison of reactive nitrogen deposition over China, *Atmos. Chem. Phys.*, 20, 10587–10610, 2020.
- Liu M, et al. Ammonia emission control in China would mitigate haze pollution and nitrogen deposition, but worsen acid rain. *Proceedings of the National Academy of Sciences* 116, 7760 (2019).
- Sadiq, M., et al. (2017). "Effects of ozone–vegetation coupling on surface ozone air quality via biogeochemical and meteorological feedbacks." *Atmospheric Chemistry and Physics* 17(4): 3055-3066.
- Zhao, Y., et al. (2017). "Responses of surface ozone air quality to anthropogenic nitrogen deposition in the Northern Hemisphere." *Atmospheric Chemistry and Physics* 17(16): 9781-9796.
- Zhou SS, Tai APK, Sun S, Sadiq M, Heald CL, Geddes JA. Coupling between surface ozone and leaf area index in a chemical transport model: strength of feedback and implications for ozone air quality and vegetation health. *Atmos Chem Phys* 18, 14133-14148 (2018).

Wang T, Lin X, Peng S, Cong N, Piao S. Multimodel projections and uncertainties of net ecosystem production in China over the twenty-first century. Chin Sci Bull 59, 4681-4691 (2014).

Response to Reviewer #3

Reviewer #3 (Remarks to the Author):

General.

This study analyses the mechanism of a non-linear response of reactive nitrogen deposition (in particular in Eastern China) to NO_x emission reductions. While the issue of non-linear responses is well known in the atmospheric chemistry community, the interesting aspect is that emission changes are so large in a relatively short time (e.g. a downward emission trend of 20 % or more), making these feedbacks more visible than would be the case under gradually changing conditions. The manuscript is interesting, but there are several loose ends that require to be corroborated, additional testing and analysis.

Response: We accepted all the reviewer's comments and revised the manuscript responsively. Please see the point-to-point responses in the following.

Specifically:

Reviewer #3: 1) Given spatial and temporal variability, the observational evidence of using only 4 deposition stations over Eastern China for 5 years is very limited. If more data is available also for earlier years, they need to be analysed as well to better understand the trends and variability from 2010-2015. One possible model simulation to demonstrate variability is keeping emissions constant, and analyse the model responses.

Response: Accepted.

- 1) We increase the number of observation stations to 10 for the 5 years over the Eastern China and surrounding areas, derived from the National Nitrogen Deposition Monitoring Network (Xu et al., 2015), representative of various land use types. The new Figure 1 with more available data illustrates that most of the stations in the Eastern China experienced relatively small reductions (<10%) or even increases in NO_y deposition despite the notable reductions (~20%) in both the NO_x emissions and NO₂ column concentrations there. Please see Figure 1 in the revised manuscript.
- 2) Following the reviewer's suggestion, we perform five-year simulations with the constant emissions based on 2015 but varying meteorological conditions from 2011–2015. The model results demonstrate that meteorological factors gave rise to diverse temporal variabilities in N deposition and could lead to concurrent positive and negative trends among different stations. Only meteorological variations cannot explain the weakened responses of NO_y deposition. We find that the chemical production of nitric acids driven by NO_x emission reductions is a key factor in driving NO_y deposition variations. Please see Line 124-135 in the revised manuscript.

Reviewer #3: 2) There is an interesting discrepancy between NO₂ column observations and the anthropogenic emissions trends (e.g. Figure 1). This needs to be explored further along with an calculation on how representative NO₂ is for NO_y dry deposition, and how it is changing. This is relevant for emerging studies that use satellite observations to compute deposition of NO_y (and NH_x).

Response: Accepted. Our model results show that in response to a 30% reduction of NO_x emissions, NO₂ columns decreased by 33%, while NO_y (including NO_x and its oxidation products like nitric acids) dry deposition decreased by less than 20% for the Eastern China because enhanced nitric acids deposition fluxes largely offset the reduction of NO_x deposition. So we suggest that using NO₂ columns observed by satellites to derive total NO_y deposition could bias the tendencies of inter-annual deposition, particularly for the Eastern China. Please see Figure 2a, Line 148-158, and Line 397-404 in the revised manuscript.

Reviewer #3: 3) Why hasn't NH₃ satellite column observations not been used for analysis (given the probably important role in buffering some of the excess nitrate produced).

Response: Accepted. The NH₃ satellite column observations are added in Figure 1. It shows that the annual mean NH₃ columns increased markedly from 2011 to 2015 in the Eastern China, consistent with the results in our previous study (Liu et al., 2018). The increased availability of free NH₃ could facilitate the partitioning of nitrate toward the aerosol phase and would consequently reduce nitrate dry deposition, in contrary to the stabilized NO_y deposition observed in the Eastern China. So NH₃ availability is the not primary cause of NO_y deposition variation. Please see revised Figure 1.

Reviewer #3: 4) The role of atmospheric transport in feedback on deposition between regions has not been discussed. Perturbations similar to MICS-Asia, HTAP etc, where only emissions over East China, South China and the rest of China, would be helpful to explore transport feedbacks

Response: Accepted. We consider the role of atmospheric transport in N deposition by using the additional simulations with the emissions from the Eastern China, the Southern China, and the rest of China excluded separately. The results show that the local emissions (of all anthropogenic sources) in the Eastern China dominate (57%) the total NO_y deposited there, with the remaining part (43%) contributed by the NO_x emissions from the other domestic regions. The emissions in the Eastern China also account for 26% of NO_y deposition in the Southern China. These results suggest that the transboundary transport of NO_y species can significantly affect the regional N deposition. We also demonstrate that the year-to-year variations of atmospheric transport do not affect our major findings. Please see the results in Line 90-108 and Line 161-167 in the revised manuscript.

Reviewer #3: 5) Appropriate budget analysis possibly combined with point 4, would be helpful to understand the closure of emissions, chemistry, transport and removal. An analysis of overall NO_y lifetime (for China, and its subregions) would be helpful as well.

Response: Accepted.

1) We calculate the ratios of the deposition to the emission for China and the sub-regions. Our simulations reveal that about half of the domestic emissions are transported out of China, consistent with those results (around 50%) estimated in previous modelling studies (Zhao et al., 2017; Ge et al., 2020). It has also been shown in the previous comment that the NO_x emissions over the Eastern China contribute substantially to the NO_y deposition in the surrounding areas.

- 2) The overall NO_y lifetime is determined by using the ratio of the simulated NO_y column concentrations to its deposition at regional scales. We find that the accelerated conversion of NO_x to HNO_3 results in a higher deposition rate of NO_y and a reduction (from 4.5 to 4.1 days) of NO_y lifetime over the Eastern China, while the NO_y lifetimes are similar between the Base and the 30% NO_x reduction cases for both the whole China and the Southern China.

Please see these revisions in the Line 90-108 and Line 198-214 in the revised manuscript.

Reviewer #3: 6) In view of point 5, an appropriate analysis of the chemical budget terms (in particular all the loss reactions NO_2+OH ; NO_2+O_3 ; N_2O_5 heterogenous) would be useful. The papers main hypothesis is that due to a shift of NO to NO_2 (NO_x component), and also more OH and other oxidants, the loss of NO_y is speeding up. This can only work if subsequently the HNO_3 is not forming aerosol nitrate. The paper is ignoring an analysis of this and the analysis of the nitrate (HNO_3 , and NH_4NO_3 and other nitrates) must be shown.

Response: Accepted.

- 1) The NO_x removal by the photochemical reactions (NO_2+OH) and nocturnal reactions (NO_2+O_3 and N_2O_5 formation) are analyzed in this study. In line with the recent studies (Wang et al., 2017; Chen et al., 2020; Fu et al., 2020), our simulations show that the gas-phase oxidation of NO_2 by OH radical and nighttime reactions mediated by NO_3 radical are major two pathways of nitric acid formation in wintertime with relative contributions of 40% vs. 60%, respectively, while the photochemistry pathway is dominant (90%) in summer. Importantly, this study finds that the enhanced chemical formation of nitric acids by NO_x reductions is contributed by both the daytime (43%) and the nighttime (57%) reactions during wintertime.
- 2) A detailed analysis of the nitrate partitioning between gas and aerosol phases is given. The simulations with the reductions of NO_x emissions show that about 50% of nitric acid presented in the gas phase, as ammonia is not enough to neutralize them completely to form ammonium nitrate. This is supported by the evidence that the fractions of particulate nitrate in the total nitrate (gas + particles) could be significantly elevated by increased ammonia availability in eastern China in our previous study (Liu et al., 2018) and Zhai et al. (2021). Therefore, the deposition rates of NO_y turn to be higher due to increased gas-phase nitrate.

Please see all these revisions in Line 116-121, Line 184-197, and Line 215-234 in the revised manuscript.

Reviewer #3: 7) At several places in the publication the word effectiveness needs to be replaced, possibly by relative response or similar.

Response: Accepted. We replace most of them throughout the manuscript.

Reviewer #3:

I do think the work can be of interest to Nature if my major comments above, and detailed comments below are duly taking into account.

Response: Accepted. We fully addressed these comments. Please see the following responses.

Reviewer #3: Detailed comments

l. 34 clarify largest in budget terms- or in terms of deposition fluxes?

Response: Accepted. We reword this sentence as: Terrestrial ecosystems in China receive the world's largest amount of reactive nitrogen (N) deposition. Please see Line 33-34 in the revised manuscript.

l. 37 It is not clear what is meant with 'integrating a model with observation'- I think the authors mostly compare (with the exception of dry dep, that is using modelled deposition velocities. Please clarify.

Response: Accepted. The national observations are used to indicate the inter-annual tendency of N deposition fluxes and to validate the model. We reword this sentence as: By combining the observations with a chemical-transport model. Please see Line 36-37 in the revised text.

l. 39 what was expected on the basis of what? The word expect suggests that an initial evaluation was made. I suggest 'unforeseen' as more appropriate. In comparing USA, Southern and Eastern China, it should be discussed that they have a very different geographical extent and make up of emissions within the regions, which makes such numbers somewhat comparing apples and pears.

Response: Accepted. We replace the word 'lower than expected' by 'unforeseen' in this sentence. We made the comparison of N emission-deposition responses among USA, Southern and Eastern China because all are source regions of NO_x and other precursors, where large emissions changes would considerably affect NO_y deposition. Please see Line 37-40 and Line 333-344 in the revised text.

l. 43-44 should be considered. I think most if not all models would include such feedbacks through atmospheric chemistry, but I leave it open whether these models have the correct response due to issues with model resolution, transport, chemistry, emissions. It is likely that the model study presented here, was more accurate than e.g used in MICS Asia, HTAP exercises, but that remains to be demonstrated.

Response: Accepted. We remove this sentence. The unexpected N emission-deposition response is demonstrated explicitly for the first time, while some other models in MICS-Asia and HTAP may be capable to consider such feedbacks. Importantly, we provide more results and analysis on ecosystem impacts based on a land model, as suggested by another reviewer.

l. 54 I recommend to use consistently in this study the units NO_x-N and NH₃-N for emissions. Currently it is not clear what this, most like NO_x expressed as NO₂. If the authors want to use these units, the unit need to be 30 Tg Nox(NO₂). yr⁻¹.

Response: Accepted. In line with the deposition values mentioned in this study, we express the unit of emissions as Tg N yr⁻¹ for NO_x and NH₃. Please see Line 53 in the revised manuscript.

l. 57 Indeed the Qian paper report stagnant emissions for NH₃, but I recommend to be somewhat careful with this, as that paper doesn't provide much information on what basis this assertion was made. A rough cross check could be made by comparing to trends in fertilizer production and imports and legumes import as a proxy for activity data that are ultimately causing NH₃ emissions

(in the absence of changing management and controls). As this could possibly change the outcomes of the study, it would be good to pay more attention to this aspect.

Response: Accepted. Our recent works (Liu et al., 2018; 2019) has clearly demonstrated that NH₃ emissions in China keep almost constant from 2011–2016, as a result of slightly decreased agricultural production and increased non-agricultural activities. Please note that the paper (Qian) mentioned by the reviewer was not cited in our manuscript.

l. 69 nitric acid and (neutralized) aerosol nitrate and organic nitrates. List is not complete

Response: Accepted. These components are included in the sentence. Please see Line 71-72.

l. 70 OH react with NO₂ (not with NO_x in general). O₃ reacts with NO to form NO₂. O₃ reacts with NO₂ to form NO₃, which under light conditions is rapidly photolysed. Please be more exact.

Response: Accepted. The sentence is reworded. We describe the primary pathways for the formation of nitric acids under both daytime and nighttime conditions. Please see Line 72-75 in the revised text.

l. 89 See comment on figure S1.

Response: Accepted. The revisions are made for Figure S1.

l. 92 social-economic=>just economic is sufficient in this context

Response: Accepted. We reword this sentence. Please see Line 95-97 in the revised text.

l. 96 so NH₃-N deposition 5.8 Tg N? Please mention this explicitly. Comparing to the earlier mentioned NO_x-N emission of 9.13 and NH₃-N emissions of 8.2 Tg N, a back of the envelop calculation gives 57 % of NO_x deposited on land- and 63.5 % of ammonia N? A budget analysis (including changes could be developed as strong key message of this paper) as the fraction remaining on land.

Response: Accepted.

- 1) We present the exact amount of NH_x-N deposition (5.8 Tg N yr⁻¹) in this sentence and also note it in Table S1.
- 2) Yes. Our simulations show that 50% of emitted NO_x (10.3 Tg N yr⁻¹) from domestic sources and 63% of NH_x-N emissions (8.2 Tg N yr⁻¹) were deposited on Chinese land, similar to the results in previous modelling studies (like Zhao et al., 2017; MICS-Asia by Ge et al., 2020). Please see Line 102-108.
- 3) Reductions of NO_x emissions increased the fraction of NO_y deposited to the Eastern China due to increased overall deposition rate of NO_y, as reflected by the increased ratio of NO_y deposition to the NO_x emissions in the region by 16%. That's why the relative response of N deposition to NO_x emission controls is appreciably lower than 100%. Please Line 198-214.

l. 97 here the authors should be more specific on the form of NO₃: as gaseous HNO₃ or in the form of aerosol nitrate? The specific form influences deposition strongly and we need to know whether this is part of the response function.

Response: Accepted. We clarify that the expression as NO₃-N in our simulations is the sum of N presented as gaseous and particulate nitrate species and dinitrogen pentoxide.

l. 103 see comments to Figure S2

Response: Accepted. We add statistical indexes for the observation-model comparison along with the scatter plots.

l. 107 a substantial change of NO_y lifetime will depend on the ability to form aerosol ammonium (and other) nitrates. It is not clear what is compared in S2 (gas, aerosol, both?) and I think it is not nitrite/nitrous acid as erroneously in S2.

Response: Accepted. We clarify that Figure S2 presented the comparison between the simulated aerosol nitrate concentrations and observations at in-situ stations over China. The observations for nitrite or nitrous acid were not available.

l. 109. Which year is 'now'?

Response: Accepted. We reword this sentence. Please see Line 138-140.

l. 112 what is the baseline case? Emissions constant or fluctuating over the years? Clarify shortly here, and methods for more details.

Response: Accepted. The definition of the baseline case was given in the main text. We clarify again in this sentence. It was developed using the emissions and meteorological inputs for the year 2015. Please also see Table S1 for the description of all simulation cases.

l. 116 see comment to Figure 2. The paper really needs to explore better the role of change in residence time and the role of atmospheric transport, including monsoon patterns.

Response: Accepted.

- 1) The lifetime of NO_y is estimated using the NO_y column concentrations and deposition at regional scales. We find that the enhanced chemical conversion of NO_x to HNO₃ in the Eastern China by NO_x reductions results in a larger deposition rate of NO_y due to a much larger deposition rate of gaseous HNO₃ than NO_x species, and consequently a reduced lifetime of NO_y from 4.6 days to 4.1 days. These results demonstrate why the relative response of NO_y deposition to the NO_x emission change is less than 70% in the region. Please see Line 198-214 in the revised manuscript.
- 2) Though atmospheric transport of N compounds is important for regional N deposition, our results for the response of N deposition to NO_x emission changes are robust despite the variations of atmospheric transport including monsoon circulation between 2011 and 2015 (Li et al., 2016; Zhang et al., 2019). Please see 161-167 in the revised manuscript.

l. 119 I think effectiveness is not the correct word- response ratio is more appropriate.

Response: Accepted. We replace most of them by the word ‘relative response’ throughout the manuscript. The word effectiveness has been used by Tan et al., 2020 to represent the ratio of relative changes in N deposition to the relative changes in its emission.

l. 122 this would point to more NO₃ in the form of aerosol nitrate, which would work to prolong lifetime?

Response: We clarify that such increased NO₃-N resides largely in the gas phase, pointing to decreased lifetime of NO_y due to a larger deposition rate of gas-phase nitrate. The result combining with the following analysis suggests that the increased production rate of NO₃-N due to enhanced oxidation capacity relieves the reductions of the regional NO₃-N deposition. Please see Line 184-197 in the revised text.

l. 143 what is present?

Response: Accepted. We have deleted this sentence.

l. 166 A thorough chemical budget analysis would pinpoint these hypothesis.

Response: Accepted. We show more quantitative results on how different chemical mechanisms (photochemical reactions and nocturnal reactions) play roles in the enhanced formation of nitric acids and the subsequent N deposition. Please see Line 198-214 and Line 215-234 in the revised text.

l. 333 this is the response to the earlier question (single year of emission or variable), please summarize in earlier sentence.

Response: Accepted. We describe here and in earlier sentences that the baseline simulation was performed using the meteorological reanalysis data and the emission for the year 2015. Please see Table S1 for more details.

l. 357 what is measured. Aerosol nitrate, gaseous nitric acid or both together. Duly discuss possible sampling errors.

Response: Accepted. The deposition fluxes of gaseous and aerosol nitrate were measured together at those in-situ stations. Possible sampling errors are provided in the cited references (Xu et al., 2015; Xu et al., 2019). Please see Line 450-457 in the revised manuscript.

l. 365 Why was not a similar comparison with NH₃ made?

Response: Accepted. The NH₃ observations are given in Fig. 1 along with NO₂ observations and emissions.

Fig 1: Please define coordinates of ‘Eastern China’ or refer to Figure 2. It would also be interesting to present the domain integrated deposition fluxes, possibly in the same units Tg NO_y-N, and NH₃-N. It is not clear why only 4 stations were selected.

Response: Accepted. We refer to Fig. 2 for the location of Eastern China. The domain averaged deposition fluxes were not employed because they could mask the trends of those stations with relatively small amounts. Deposition fluxes differed by up to a factor of five among the station. So we display as many stations as possible to reflect the long-term variations at regional scales. Please see Figs. 1 and 2 for more details in the revised manuscript.

Fig 2: Is this a 4 year average? Does panel a) include Eastern China. It would probably be more instructive to have a separate panel for 'other China'. What exactly is displayed in 2c? I think it is a scaled response to 30 % emission reduction, not an effectiveness, as part of the response can be due to transport. Also it took me until figure 3 to understand that the green colors over Eastern China are actually higher deposition. Probably it makes sense to first Figure 3 and then 2? Or add a sentence in the figure caption guiding the readers' interpretation.

Response: Accepted. We revise the manuscript as following:

- 1) The resulted for the Eastern China and the Southern China are presented separately in the panels. Please see revised Figure 2.
- 2) As the reviewer suggested, it could be more understandable to use the word response than to use effectiveness. The figure and caption are therefore revised. While atmospheric transport is important for regional N deposition, those response changes of N deposition are the results of the overall reduction (local + regional) of NO_x emissions in China. In Figure 2, the meaning of color scales is explained in the figure caption to guide readers.

Fig 3: the enhanced response of NO₃ deposition over Eastern China, in particular in winter, raises question what the chemical mechanism really is. Conventional wisdom points to heterogeneous (N₂O₅ possibly also NO₃ radical) reactions as the dominant driver of NO_x removal, but it is not clear why these reaction rates would be enhanced in winter. Unfortunately, the paper doesn't do a thorough job in analyzing these chemical mechanism when attempting to understand the drivers of these responses, although there may be clues in Figure 4, where in particular NO₃ radical seems to have a strong response. More analysis needed.

Response: Accepted. We perform the sensitivity simulations with the nighttime pathway turning on and off. The results reveal how different chemical mechanisms of NO_x removal (i.e., NO₂+OH and NO₃+NO₂) play roles in the enhanced formation of nitric acids and the subsequent N deposition. Please see Line 116-123 and Line 215-234 in the revised manuscript.

Figure 4: You should also plot (numerically) the absolute values of O₃, NO₃ and OH, to get a feeling of the importance of the numbers.

Response: Accepted. Their absolute changes due to NO_x emission reductions are added in Figure 4 in conjunction with the relative variations. Please see revised Figure 4.

Figure 4: What is going on over Tibet with ratios close to zero "Black stars in figure 2c refer to the four observations in Figure 1?"

Response: Accepted. It appears that this question sourced from the results in Figure 2c (not Figure 4) showing the ratio of deposition changes to the emission changes. The ratios were very small

over Tibet because only anthropogenic NO_x emissions over China were altered in our simulations and the resulting changes in N deposition for the pristine regions like Tibet were negligible. The black stars in Figure 2c denote the locations of the ten observation stations used in Figure 1.

Fig S1 These are nice plots, but it is difficult to understand the model skills. I recommend to include 2 additional panels that provide a scatter plot+some spatial skill statistics. Also to exclude variability issues it would be better to evaluate skills over a 5 year period rather than a single year.

Response: Accepted. A scatter plot for the model-observation comparison and the statistics indexes are added. The model variability over this period is discussed in our manuscript. Please see revised Figure S1 and Line 124-135 and Line 161-167 in the revised manuscript.

Figure S2 Probably the authors meant nitrate (not nitrite). The way this is presented suggest better model performance. A commonly used statistic is also the fraction of model results with a factor of 0.5 and 2 from observations.

Response: Accepted. The performance of simulated nitrate is evaluated with available observation data. The statistics calculation for the fraction of model results with a factor of 0.5 and 2 from observations is added. It shows most of the observation-model biases within a factor of 2. Please see revised Figure S2.

References:

- Chen X, et al. Field Determination of Nitrate Formation Pathway in Winter Beijing. *Environ Sci Technol* 54, 9243-9253 (2020).
- Fu X, et al. Persistent Heavy Winter Nitrate Pollution Driven by Increased Photochemical Oxidants in Northern China. *Environ Sci Technol* 54, 3881-3889 (2020).
- Ge B, et al. Model Inter-Comparison Study for Asia (MICS-Asia) phase III: multimodel comparison of reactive nitrogen deposition over China. *Atmos Chem Phys* 20, 10587-10610 (2020).
- Li Q, Zhang R, Wang Y. Interannual variation of the wintertime fog-haze days across central and eastern China and its relation with East Asian winter monsoon. *Int J Climatol* 36, 346-354 (2016).
- Liu M, et al. Rapid SO₂ emission reductions significantly increase tropospheric ammonia concentrations over the North China Plain. *Atmos Chem Phys* 18, 17933-17943 (2018).
- Liu M, et al. Ammonia emission control in China would mitigate haze pollution and nitrogen deposition, but worsen acid rain. *Proceedings of the National Academy of Sciences* 116, 7760 (2019).
- Tan J, Fu JS, Seinfeld JH. Ammonia emission abatement does not fully control reduced forms of nitrogen deposition. *Proc Natl Acad Sci U S A* 117, 9771-9775 (2020).
- Xu W, et al. Quantifying atmospheric nitrogen deposition through a nationwide monitoring network across China. *Atmos Chem Phys* 15, 12345-12360 (2015).
- Xu W, Zhang L, Liu X. A database of atmospheric nitrogen concentration and deposition from the nationwide monitoring network in China. *Sci Data* 6, 51 (2019).

- Zhai S, et al. Control of particulate nitrate air pollution in China. *Nat Geosci* 14, 389-395 (2021).
- Zhang G, et al. Seesaw haze pollution in North China modulated by the sub-seasonal variability of atmospheric circulation. *Atmos Chem Phys* 19, 565-576 (2019).
- Zhao Y, et al. Atmospheric nitrogen deposition to China: A model analysis on nitrogen budget and critical load exceedance. *Atmos Environ* 153, 32-40 (2017).

REVIEWER COMMENTS

Reviewer #1 (Remarks to the Author):

Referee comments for NCOMMS-21-21037A-Z: “Unexpected response of nitrogen deposition to nitrogen oxide controls and implications for land carbon sink”

The authors took the reviewer's comments into full consideration and revised the manuscript thoroughly. The paper is more clear and improved than previous version. However, I will not recommend the authors to discuss the effect of N deposition on land carbon sink in this paper, which make the topic of this study seems too big and unfocused. Of cause, the effect of N deposition on carbon balance is an interesting topic, but not the key point of this study. Furthermore, the regional simulation of earth system model or dynamic vegetation model also need multi-source input data, parameter localization, and data validation. A preliminary modelling analysis would bring large uncertainty in the estimation for the effects of declined N deposition on carbon sequestration.

For example, what is the land use and land cover data used in the simulation? Did you consider the effect of CO₂, land use and land cover change, and N fertilization on NEP in the model? Did you compare the modeling NEP results with the observation results from eddy Flux tower?

Line55-56 : this reference seems rather old (Tian et al., good paper but not up to date).

Line73-75: unclear. Please clarify.

Reviewer #2 (Remarks to the Author):

The authors have made substantial revisions to the manuscript that have mostly addressed our concerns in an adequate and reasonable manner. In the light of some more recently published work, I recommend however that some more discussion should be warranted before this paper should be accepted. Please see below for details.

In a recent study, Liu et al. (2021) (<https://doi.org/10.5194/acp-21-17743-2021>) showed that the vegetation responses to changes in nitrogen deposition in China are relatively small, because most of

China's terrestrial ecosystems are not very nitrogen-limited. This paper, however, appears to show greater sensitivity of vegetation to changing nitrogen deposition. What could be the sources of differences? Would it have to do with, say, difference in the model used (Liu et al. used CLM4.5 while the authors here used CLM5), or the different treatment of nitrogen limitation as well as nitrogen biogeochemistry in general? That Liu et al. examined rising nitrogen deposition in the light of rising agricultural production, whereas the authors here examined decreasing nitrogen deposition due to emission control, and thus the sensitivity may be nonlinear? A more thorough comparison of this work with Liu et al. (2021), dissecting the differences and similarities, on a par with the other previous related work (e.g., Zhou et al., (2017)) is necessary for a compelling paper to be published.

Reviewer #3 (Remarks to the Author):

The authors have largely adequately responded to my previous concerns. This is an interesting research paper, which shows that atmospheric chemistry effects can lead to non-linear responses to emission reductions in China, with unforeseen consequences for climate mitigation objectives. The additional carbon cycle analysis, is a valuable addition to the paper, but may need some additional discussion to cover all possible consequences. Although the English used in the manuscript is in general of a quite good level, but some additional editing would be helpful to remove the last glitches. This also applies to at some places, sloppy use of language that may be somewhat confusing to the reader.

All in all, I recommend this manuscript now for publication in Nature Communications, and I add below some minor suggestions for further improvement of the manuscript.

Minor suggestions>:

I. 36 Thanks for revising this wording- here is a suggestion that I think covers better what you did: "Using a chemical transport model to understand observed spatial and temporal deposition trends, we show that ... etc"

I. 42 NO_x reductions =>"No_x emission reductions"

I. 42-43 This is a new part of the paper (and requested by one of the reviewers): to me it is not a priori clear why this is a penalty. It is probably related to relatively less nitrogen transport to regions where more nitrogen deposition could lead to an enhanced nitrogen sink.

I. 56 Suggest to use terrestrial instead of natural.

I. 60 explain what is meant by passed decade (2011-2021??).

I. 90-100. The model observation correspondence displayed in Fig S1-S2 is objectively not great- and probably cosmetically looks somewhat better by the use of log-log scatter plots. Rather than claiming that this 'generally agreed', I suggest to focus on the given the arguments why this is fit-for-purpose, or is state-of-the-art. My expectation is that this model performance is quite dependent on the specifics of the observational network and the complexities of the pollution chemistry in China. I do suggest to have some additional analysis along with S1/S2 (and not in the main text).

I. 98 mention here what method you used to derive the the relative contribution to deposition in East and South Asia.

I. 108 Great. These results collectively suggest that the base simulation of WRF-CHEM is in line with literature knowledge on large scale transport process within China and between China and surrounding regions.

I. 109 NO₃ is aerosol nitrate (NO₃⁻)+HNO₃ and N₂O₅ I think.

I. 140 *emissions* of other species constant.

L 142 average for the areas defined in 2c?

I. 144 Nationally=>Nationally aggregated?

I. 290: language. Do you mean enhance the efficacy? Suggest the additional VOC emission control led to a more linear response of NO_y deposition to NO_x emission controls

I. 306 I like the additional carbon cycle simulations, but I also notice that they are not sufficiently described and discussed. I understand from the methods section that a 30 years simulations was performed under near-equilibrium conditions. However, the situation is of course far from equilibrium, and something needs to be said what the likely response to transient N-dep trajectories will be- either by adding 1 additional scenario, or by discussing some literature result. As the emission-deposition responses have a strong regional signature, I expect also some discussion on the covariation of ecosystem location and deposition changes. For instance if for some other reason it will still be attractive to preferentially reduce NO_x emissions (without VOC), would a possible response be to also favor reforestation in Eastern China? Maybe this is not a very realistic suggestion, but something needs to said about it (in discussion?).

Figures:

Figure 1: it is it possible to provide in the figure information where the stations are located, as this is important to understand the 'story'. Likewise, can the station codes be added to figure 2 panel c? Also somewhere the station coordinates should be listed.

Figure 2c: for which scenario (10/30/50) the results in 1c apply? mention in caption

Supplement:

Figure S2: although the authors claim to have corrected nitrite to nitrate, the error is still present.

Response to reviewers' comments

Reviewer #1 (Remarks to the Author):

(comments are marked in italic)

Referee comments for NCOMMS-21-21037A-Z: "Unexpected response of nitrogen deposition to nitrogen oxide controls and implications for land carbon sink"

Reviewer #1: The authors took the reviewer's comments into full consideration and revised the manuscript thoroughly. The paper is more clear and improved than previous version. However, I will not recommend the authors to discuss the effect of N deposition on land carbon sink in this paper, which make the topic of this study seems too big and unfocused. Of course, the effect of N deposition on carbon balance is an interesting topic, but not the key point of this study.

Response: We appreciate the reviewer's comment. The topic of this study was to unravel the mechanisms in the weakened response of N deposition to nitrogen oxide emission controls in China. The vast majority of our data analysis and the conclusion were focused on that. Interestingly, our study further demonstrates that the N deposition changes in response to emission controls can appreciably modulate ecosystem production (Please see Line 317-328 and 343-365 in the revised manuscript). This implication is strongly recommended by other two reviewers.

Reviewer #1: Furthermore, the regional simulation of earth system model or dynamic vegetation model also need multi-source input data, parameter localization, and data validation. A preliminary modelling analysis would bring large uncertainty in the estimation for the effects of declined N deposition on carbon sequestration.

For example, what is the land use and land cover data used in the simulation? Did you consider the effect of CO₂, land use and land cover change, and N fertilization on NEP in the model? Did you compare the modeling NEP results with the observation results from eddy Flux tower?

Response: Accepted. In the revision, we followed CMIP6 protocol to rerun CLM5 in a transient period between 1850 and 2014 and updated the results with such comprehensive simulation (Please see Line 500-508). We provided more details of the input data and model configuration in our land model simulations:

- 1) We introduced the use of transient land use change data and described the scheme of nitrogen use in both plant and soil microorganisms. We mentioned how photosynthetic rates respond to the elevating CO₂ concentration in the Farquhar photosynthesis model. Especially, we explain how fertilized N could benefit plant from growth to gain carbon. (Please see Line 481-499)
- 2) As the land component of the community earth system model (CESM2.0), CLM5 was used in CMIP6 as the reference of IPCC AR6 report. CLM5 carbon fluxes in both global scale and site scale including China have been systematically evaluated

in the ILAMB project (e.g., Collier et al., 2018; Lawrence et al., 2019) (Please see Line 495-499).

- 3) Our team has been highly involved in the development of CLM. Dr. Yongjiu Dai, as one of the coauthors in this manuscript, has initialized the development of CLM (Dai et al., 2003). Another coauthor, Dr. Xingjie Lu, has recently improved the representation of carbon and nitrogen cycles in CLM5 simulations (Lu et al., 2020). So we believe, with our revision, results of CLM5 would provide important reference for future estimates on land carbon sinks.

Reviewer #1: Line55-56: this reference seems rather old (Tian et al., good paper but not up to date).

Response: Accepted. We updated the references here by including Zhu et al., 2021. Please see Line 58-59.

Reviewer #1: Line73-75: unclear. Please clarify.

Response: Accepted. These sentences were reworded as: The daytime photo-oxidation of NO₂ by OH radical and the nighttime NO₃ radical-involved chemistry are the major pathways for the formation of NO₃-N compounds in the lower troposphere. Volatile Organic Compounds (VOCs) can readily react with OH radical and promote O₃ formation. Please see Line 76-80.

Reference:

Collier, N. et al. The International Land Model Benchmarking (ILAMB) System: Design, Theory, and Implementation. *J. Adv. Model. Earth Syst.* 10, 2731-2754 (2018).

Dai, Y. et al. The common land model. *Bulletin of the American Meteorological Society*, 84, 1013-1024 (2003).

Lawrence, D. M. et al. The Community Land Model Version 5: Description of New Features, Benchmarking, and Impact of Forcing Uncertainty. *J. Adv. Model. Earth Syst.* 11, 4245-4287 (2019).

Lu, X. et al. Full implementation of matrix approach to biogeochemistry module of CLM5. *Journal of Advances in Modeling Earth Systems*, 12, e2020MS002105 (2020). <https://doi.org/10.1029/2020MS002105>.

Zhu J, Wang Q, He N, Yu G. Effect of atmospheric nitrogen deposition and its components on carbon flux in terrestrial ecosystems in China. *Environ. Res.*, 202, 111787 (2021).

Reviewer #2 (Remarks to the Author):

The authors have made substantial revisions to the manuscript that have mostly addressed our concerns in an adequate and reasonable manner. In the light of some more recently published work, I recommend however that some more discussion should be warranted before this paper should be accepted. Please see below for details.

In a recent study, Liu et al. (2021) (<https://doi.org/10.5194/acp-21-17743-2021>) showed that the vegetation responses to changes in nitrogen deposition in China are relatively small, because most of China's terrestrial ecosystems are not very nitrogen-limited. This paper, however, appears to show greater sensitivity of vegetation to changing nitrogen deposition. What could be the sources of differences? Would it have to do with, say, difference in the model used (Liu et al. used CLM4.5 while the authors here used CLM5), or the different treatment of nitrogen limitation as well as nitrogen biogeochemistry in general? That Liu et al. examined rising nitrogen deposition in the light of rising agricultural production, whereas the authors here examined decreasing nitrogen deposition due to emission control, and thus the sensitivity may be nonlinear? A more thorough comparison of this work with Liu et al. (2021), dissecting the differences and similarities, on a par with the other previous related work (e.g., Zhou et al., (2017)) is necessary for a compelling paper to be published.

Response: Accepted. We provide more discussions of our land model results with the context of other related studies (e.g., Liu et al., 2021; Zhao et al., 2017). As noted by the reviewer, the comparison of this work with Liu et al. (2021) includes the following points:

- 1) Difference in the model version of CLM. There have been substantial updates of nitrogen cycle modules in CLM5.0 compared to CLM4.5. One of the most significant updates in CLM5.0 nitrogen cycle model is to incorporate Fixation and Uptake of Nitrogen model (FUN). This update considers the carbon cost of plant nitrogen acquisition such that net primary production (NPP) downregulates further to the nitrogen limitation. As a result, vegetation carbon uptake in CLM5 may show greater sensitivity to the nitrogen deposition changes. (Please see Line 489-498)
- 2) Difference in the emission scenarios. We applied a 30% reduction of anthropogenic NO_x emissions across China, resulting in a 12% reduction of the total nitrogen (sum of reduced and oxidized nitrogen) deposition nationwide; while Liu et al. (2021) considered the rising nitrogen deposition by 5-15% in northern China and less than 5% in southern China. The decline in N fluxes may enhance nitrogen limitation. Hence, our modeling cases tend to produce larger changes in ecosystem production.
- 3) The regional contrast of N deposition responses is in line with previous related works. We show that the fertilization effect of nitrogen deposition was much more significant in southern China than in northern China (Fig. 4). Similarly, Liu et al. (2021) find a reduction of gross primary production mostly in southern China due to nitrogen limitation; Zhao et al. (2017) show marked contributions (5-30%) of anthropogenic nitrogen deposition to NPP and leaf area index in southern China. Such discussion is included in the main text. (Please see Line 343-355)

References:

Liu X, Tai APK, Fung KM. Responses of surface ozone to future agricultural ammonia emissions and subsequent nitrogen deposition through terrestrial ecosystem changes. *Atmos Chem Phys* 21, 17743-17758 (2021).

Zhao Y, Zhang L, Tai APK, Chen Y, Pan Y. Responses of surface ozone air quality to anthropogenic nitrogen deposition in the Northern Hemisphere. *Atmos Chem Phys* 17, 9781-9796 (2017).

Reviewer #3 (Remarks to the Author):

(Comments from the reviewer are marked in italic)

The authors have largely adequately responded to my previous concerns. This is an interesting research paper, which shows that atmospheric chemistry effects can lead to non-linear responses to emission reductions in China, with unforeseen consequences for climate mitigation objectives. The additional carbon cycle analysis, is a valuable addition to the paper, but may need some additional discussion to cover all possible consequences. Although the English used in the manuscript is in general of a quite good level, but some additional editing would be helpful to remove the last glitches. This also applies to at some places, sloppy use of language that may be somewhat confusing to the reader.

All in all, I recommend this manuscript now for publication in Nature Communications, and I add below some minor suggestions for further improvement of the manuscript.

Response: We appreciate the reviewer's further comments/suggestions and accept all of them to improve our manuscript. The English language has been checked and polished throughout the manuscript. Please see our point-to-point responses below.

Minor suggestions>:

l. 36 Thanks for revising this wording- here is a suggestion that I think covers better what you did: "Using a chemical transport model to understand observed spatial and temporal deposition trends, we show that ... etc"

Response: Accepted. We reword this sentence following the reviewer's suggestion. Please see Line 38-39.

l. 42 NO_x reductions => "NO_x emission reductions"

Response: Accepted. We add the word 'emission' here. Please see Line 44-45.

l. 42-43 This is a new part of the paper (and requested by one of the reviewers): to me it is not a priori clear why this is a penalty. It is probably related to relatively less nitrogen transport to regions where more nitrogen deposition could lead to an enhanced nitrogen sink.

Response: Accepted. Because nitrogen deposition can stimulate plant primary production and associated carbon sinks via the fertilization effect (e.g., Tian et al., 2011; Zhu et al., 2021), declining nitrogen deposition from anthropogenic sources in China can bear a penalty on terrestrial carbon sequestration. The related discussion is included in our manuscript. Please see Line 317-328 and 356-365.

l. 56 Suggest to use terrestrial instead of natural.

Response: Accepted. We replace it by the word 'terrestrial' throughout the manuscript. Please see Line 59.

l. 60 explain what is meant by past decade (2011-2021??).

Response: Accepted. Here it means the period of 2010s (2010-2019). The sentence is revised. Please see Line 63.

l. 90-100. The model observation correspondence displayed in Fig S1-S2 is objectively not great- and probably cosmetically looks somewhat better by the use of log-log scatter plots. Rather than claiming that this 'generally agreed', I suggest to focus on the given the arguments why this is fit-for-purpose, or is state-of-the-art. My expectation is that this model performance is quite dependent on the specifics of the observational network and the complexities of the pollution chemistry in China. I do suggest to have some additional analysis along with S1/S2 (and not in the main text).

Response: Accepted. We add detailed analysis of model performance for nitrogen deposition and nitrate pollution in China in the Supplementary Text 1 in conjunction with Supplementary Figs. 1-2. Considering possible influences of the availability of observational networks on the model-observation comparison, the model results were compared not only with the in-situ observations, but also with previous modeling studies to provide a high confidence of our simulations. Please see details in Supplementary file.

l. 98 mention here what method you used to derive the relative contribution to deposition in East and South Asia.

Response: Accepted. To indicate the source contributions of targeted regions to deposition, we separately switched off the emissions at each source region in the parallel simulations and compared the results between them and the Baseline case. We give a simple description here and more details in the Method section and Supplementary Table 1. Please see Line 102-104 and Line 451-456.

l. 108 Great. These results collectively suggest that the base simulation of WRF-CHEM is in line with literature knowledge on large scale transport process within China and between China and surrounding regions.

l. 109 NO₃ is aerosol nitrate (NO₃-)+HNO₃ and N₂O₅ I think.

Response: Accepted. We clarify that NO₃-N is the sum of gas and aerosol NO₃⁻ and N₂O₅. Please see Line 114.

*l. 140 *emissions* of other species constant.*

Response: Accepted. We reword the sentence as 'emissions of other species constant'. Please see Line 146-147.

l. 142 average for the areas defined in 2c?

Response: The values shown here were the ratios of the changes in region-aggregated deposition to those of region-aggregated emissions for those areas of interest in Fig.

2c. Please see Line 149-151.

l. 144 Nationally=>Nationally aggregated?

Response: Accepted. The results shown here were nationally aggregated deposition. Please see Line 150-151.

l. 290: language. Do you mean enhance the efficacy? Suggest the additional VOC emission control led to a more linear response of NO_y deposition to NO_x emission controls

Response: Accepted. We reword the phrase as ‘enhance the efficacy’ in this sentence. Please see Line 291.

l. 306 I like the additional carbon cycle simulations, but I also notice that they are not sufficiently described and discussed. I understand from the methods section that a 30 years simulations was performed under near-equilibrium conditions. However, the situation is of course far from equilibrium, and something needs to be said what the likely response to transient N-dep trajectories will be- either by adding 1 additional scenario, or by discussing some literature result. As the emission-deposition responses have a strong regional signature, I expect also some discussion on the covariation of ecosystem location and deposition changes. For instance if for some other reason it will still be attractive to preferentially reduce NO_x emissions (without VOC), would a possible response be to also favor reforestation in Eastern China? Maybe this is not a very realistic suggestion, but something needs to said about it (in discussion?).

Response: Accepted. Based on the reviewer’s suggestion, we improve our analysis on the carbon cycle simulations as following:

We update the modeling results of terrestrial carbon balance from transient simulations in the period of 1850–2014 that incorporate transient nitrogen deposition data from CMIP6 model. Three scenarios using different nitrogen deposition data (baseline N deposition, reduced N deposition, and ideally-reduced N deposition) were applied for the period of present day (2005–2014). By evaluating the magnitude and the spatial patterns of land carbon sinks among different nitrogen deposition scenarios, we found that the updated results in transient simulations fully support our conclusions that have been derived from original near-equilibrium simulation. Therefore, we replace original equilibrium results with new transient results. (Please see Line 500-514)

We also agree that the effects of emission changes on ecosystem production were regionally dependent. The non-linear response of N deposition to NO_x emission controls in Eastern China could feed back to ecosystem carbon sinks. Discussion on changing nitrogen fertilization effect on plant growth is included in the text. (Please see Line 317-328 and Line 343-355)

Figures:

Figure 1: it is it possible to provide in the figure information where the stations are located, as this is important to understand the ‘story’. Likewise, can the station codes be added to figure 2 panel c? Also somewhere the station coordinates should be listed.

Response: Accepted. The locations of the stations are marked in Fig. 2c.

Figure 2c: for which scenario (10/30/50) the results in 1c apply? mention in caption

Response: Accepted. Figure 2c presents the results in the 30% reduction case. It is mentioned in the caption.

Supplement:

Figure S2: although the authors claim to have corrected nitrite to nitrate, the error is still present.

Response: Accepted. We did correct it this time. Thanks for the reminder.

Reference:

Tian H, et al. China's terrestrial carbon balance: Contributions from multiple global change factors. *Global Biogeochem. Cycles* 25, GB1007 (2011).

Zhu J, Wang Q, He N, Yu G. Effect of atmospheric nitrogen deposition and its components on carbon flux in terrestrial ecosystems in China. *Environ. Res.*, 202, 111787 (2021).

REVIEWERS' COMMENTS

Reviewer #2 (Remarks to the Author):

The authors have addressed my additional concerns adequately and I recommend publication of the revised manuscript.

Reviewer #3 (Remarks to the Author):

I have looked at the rebuttal to my (reviewer 3) comments only, and due to time constraints did not read the ms again, nor did I look at the other reviewers rebuttals.

Generally I am happy with the implemented changes, and recommend to accept this manuscript for publication.

Some last suggestions for the revised manuscript:

1) Response: Accepted. Because nitrogen deposition can stimulate plant primary production and associated carbon sinks via the fertilization effect (e.g., Tian et al., 2011; Zhu et al., 2021), declining nitrogen deposition from anthropogenic sources in China can bear a penalty on terrestrial carbon sequestration. The related discussion is included in our manuscript. Please see Line 317-328 and 356-365.

365: cautioned against is perhaps not the intended wording. I suggest 'taken into account' is more correct. i.e. which should be taken into account when developing pathways for China's carbon neutrality before 2060.

2) Response: Accepted. We add detailed analysis of model performance for nitrogen deposition and nitrate pollution in China in the Supplementary Text 1 in conjunction with Supplementary Figs. 1-2. Considering possible influences of the availability of observational networks on the model-observation comparison, the model results were compared not only with the in-situ observations, but also with previous modeling studies to provide a high confidence of our simulations. Please see details in Supplementary file.

* thanks. I do agree that model performance looks similar to other models in the field (i.e. not great), with consistent underestimates of the estimates (possibly due to observational issues, not necessarily indicating that the model is wrong. Please correct (again) the word nitrite into nitrate.

Point-to-point response

Reviewer #2 (*Remarks to the Author*):

The authors have addressed my additional concerns adequately and I recommend publication of the revised manuscript.

Response: We appreciated the reviewer's comments which helped us improve the manuscript.

Reviewer #3 (Remarks to the Author):

Reviewer #3: I have looked at the rebuttal to my (reviewer 3) comments only, and due to time constraints did not read the ms again, nor did I look at the other reviewers' rebuttals.

Generally, I am happy with the implemented changes, and recommend to accept this manuscript for publication. Some last suggestions for the revised manuscript:

1) Response: Accepted. Because nitrogen deposition can stimulate plant primary production and associated carbon sinks via the fertilization effect (e.g., Tian et al., 2011; Zhu et al., 2021), declining nitrogen deposition from anthropogenic sources in China can bear a penalty on terrestrial carbon sequestration. The related discussion is included in our manuscript. Please see Line 317-328 and 356-365. 365: cautioned against is perhaps not the intended wording. I suggest 'taken into account' is more correct. i.e. which should be taken into account when developing pathways for China's carbon neutrality before 2060.

Response: Accepted. We used 'taken into account' here. Please see Line 363-364 or our revision below:

'which should be taken into account when developing pathways for China's carbon neutrality before 2060.'

Reviewer #3: 2) Response: Accepted. We add detailed analysis of model performance for nitrogen deposition and nitrate pollution in China in the Supplementary Text 1 in conjunction with Supplementary Figs. 1-2. Considering possible influences of the availability of observational networks on the model-observation comparison, the model results were compared not only with the in-situ observations, but also with previous modeling studies to provide a high confidence of our simulations. Please see details in Supplementary file.

** thanks. I do agree that model performance looks similar to other models in the field (i.e. not great), with consistent underestimates of the estimates (possibly due to observational issues, not necessarily indicating that the model is wrong. Please correct (again) the word nitrite into nitrate.*

Response: Accepted. We corrected this word in the caption of Supplementary Fig. 2. Please see the Supplementary file or the revision below:

'Supplementary Fig. 2 Evaluation of simulated nitrate concentration against the observations during JJA and DJF of 2015. a Spatial distribution of simulated annual mean nitrate concentration in China. b Evaluation on spatial distribution of simulated mean nitrate concentration.'